# Skeletal muscle derived Musclin protects the heart during pathological overload

Malgorzata Szaroszyk[1,26], Badder Kattih [1,2,25,26✉], Abel Martin-Garrido[2], Felix A. Trogisch [2], Gesine M. Dittrich [1,2], Andrea Grund[1,2], Aya Abouissa [2], Katja Derlin[3], Martin Meier [4], Tim Holler [5], Mortimer Korf-Klingebiel[1], Katharina Völker[6], Tania Garfias Macedo[7], Cristina Pablo Tortola[8], Michael Boschmann[8], Nora Huang[8,9], Natali Froese[1], Carolin Zwadlo[1], Mona Malek Mohammadi [2], Xiaojing Luo[10], Michael Wagner[10,11], Julio Cordero[12], Robert Geffers [13], Sandor Batkai[14], Thomas Thum [14,15,16], Nadja Bork[17], Viacheslav O. Nikolaev [17], Oliver J. Müller [18,19], Hugo A. Katus[20,21], Ali El-Armouche[10], Theresia Kraft[5], Jochen Springer[22], Gergana Dobreva [12,21], Kai C. Wollert[1,16], Jens Fielitz [8,23,24], Stephan von Haehling[7], Michaela Kuhn[6], Johann Bauersachs [1,16] & Joerg Heineke [1,2,16,21✉]

Cachexia is associated with poor prognosis in chronic heart failure patients, but the underlying mechanisms of cachexia triggered disease progression remain poorly understood. Here, we investigate whether the dysregulation of myokine expression from wasting skeletal muscle exaggerates heart failure. RNA sequencing from wasting skeletal muscles of mice with heart failure reveals a reduced expression of *Ostn*, which encodes the secreted myokine Musclin, previously implicated in the enhancement of natriuretic peptide signaling. By generating skeletal muscle specific *Ostn* knock-out and overexpressing mice, we demonstrate that reduced skeletal muscle Musclin levels exaggerate, while its overexpression in muscle attenuates cardiac dysfunction and myocardial fibrosis during pressure overload. Mechanistically, Musclin enhances the abundance of C-type natriuretic peptide (CNP), thereby promoting cardiomyocyte contractility through protein kinase A and inhibiting fibroblast activation through protein kinase G signaling. Because we also find reduced *OSTN* expression in skeletal muscle of heart failure patients, augmentation of Musclin might serve as therapeutic strategy.

*A list of author affiliations appears at the end of the paper.

Despite recent advances in therapy, mortality rates in patients suffering from chronic heart failure (CHF) remain comparable or even higher than in many forms of cancer[1]. Heart failure is a systemic disease that is complicated by co-morbidities in patients. In this regard, the existence of cardiac cachexia in CHF patients has been identified as a predictor of increased mortality[2,3]. Cardiac cachexia is defined as a complex syndrome characterized by involuntary weight loss in advanced heart failure due to reduced fat, bone, and skeletal muscle mass[4–6]. Skeletal muscle wasting, which occurs in up to 20% of CHF patients, can develop prior to full cachexia and might exist even without detectable weight loss at earlier disease states[7]. Recent studies have indicated that skeletal muscle not only enables locomotion of the body, but that it also contributes to body homeostasis and the systemic stress response via secretion of soluble proteins ("myokines")[8]. Given the strong impact of muscle wasting on mortality in heart failure patients, we hypothesized that skeletal muscle as one of the largest tissues of the body might play a pivotal role in preventing the progression of CHF by producing cardioprotective myokines. While it is known that the heart secretes factors under pathological conditions to trigger skeletal muscle wasting by endocrine inter-tissue crosstalk[9–11], the mechanisms how skeletal muscle wasting aggravates heart failure in the course of cardiac cachexia remains unclear. Musclin is a myokine that is expressed mainly in skeletal muscle and in slightly lower amounts in bone (hence also referred to as osteocrin in the literature), but not in the heart[12–18]. It has been suggested that Musclin prevents the degradation of natriuretic peptides (NP) by its competitive binding to the NP clearance receptor (NPR-C)[19]. Three natriuretic peptides (ANP, BNP, CNP) and three natriuretic peptide receptors are known (NPR-A, -B, and -C). ANP and BNP bind to NPR-A, while CNP binds to NPR-B[20]. Both are transmembrane guanylyl cyclase-coupled receptors generating cyclic GMP (cGMP) as second messenger. All natriuretic peptides bind with similar affinity to their clearance receptor NPR-C[21]. While ANP and BNP are mainly expressed in the heart by cardiomyocytes, CNP is expressed in various organs, and in the heart it is mainly derived from non-myocytes. The levels of ANP, BNP, and CNP are increased during heart failure, and especially BNP is a widely used biomarker that correlates well with disease severity[20]. Functionally, all natriuretic peptides in general protect the heart by exerting for example anti-hypertrophic, anti-fibrotic and pro-contractile effects, although the role of CNP is less clear[20]. CHF, however, is typically associated with impaired natriuretic peptide signaling, i.e. with natriuretic peptide resistance. It is suggested that this occurs, at least in part, by an enhanced internalization and degradation of natriuretic peptides via NPR-C[22]. Here we show that Musclin expression is reduced in mice and patients suffering from cardiac cachexia and that skeletal muscle-specific disruption of Musclin exaggerates the progression of heart failure in mice. Elevating Musclin levels during cardiac cachexia via AAV6-mediated skeletal muscle gene transfer, in turn, ameliorates cardiac dysfunction and myocardial tissue fibrosis. Mechanistically, we show that Musclin mainly enhances cardiomyocyte C-type natriuretic peptide (CNP)/NPR-B signaling to increase cardiomyocyte contractility and to inhibit fibroblast activation.

## Results

**Cardiac cachexia in a murine model of heart failure causes Musclin deficiency in wasting skeletal muscles.** To determine whether skeletal muscles exert a differential expression pattern of myokines (secreted proteins) during cardiac cachexia, we used chronic transverse aortic constriction (TAC) to induce long-term pressure overload as a model of heart failure with cachexia in C57Bl/6J wild-type (WT) mice (Fig. 1a). WT mice subjected to TAC for 12 weeks developed maladaptive cardiac hypertrophy (increased heart weight/tibia length, HW/TL ratio), severe pulmonary congestion (increased lung weight, LW/TL ratio) and a strongly reduced echocardiographic left ventricular ejection fraction, all indicative of advanced heart failure (Fig. 1b–d). In addition, we observed reduced quadriceps, gastrocnemius, triceps, plantaris, and soleus muscle weights as well as the development of a reduced body weight in mice 12 weeks after TAC, suggesting the existence of cachexia (Fig. 1e–j). Indeed, weighing of inguinal fat and abdominal magnetic resonance imaging (MRI) scans revealed a marked reduction of inguinal and abdominal fat and a significant reduction in autochthonal and psoas muscle mass (Fig. 1k–o). The reduced skeletal muscle weights after TAC also persisted for the majority of the analyzed muscles, even when normalization to body weight was conducted (Supplementary Fig. 1a–e). Gene-expression analysis by qPCR in different skeletal muscles showed increased expression of atrophy-related genes (*MAFbx/Atrogin-1, Trim63/MuRF-1, Foxo-1*) and the pro-inflammatory gene *TNF-α*, also known as cachectin, as additional sign of cachexia (Supplementary Fig. 1f–q). Moreover, analysis of the fiber size distribution of glycolytic IId/x and IIb fibers in histological cross-sections of quadriceps muscles showed significant increases in the number of small fibers and decreases in the number of big fibers after TAC (Fig. 1p–r), as it was previously demonstrated in a mouse model of CHF[23]. We conducted RNA sequencing (RNAseq) from quadriceps muscles 12 weeks after TAC or sham surgery. In general, our analysis revealed consistent results between the biological replicates and showed that 799 genes were upregulated and 872 genes downregulated in quadriceps muscle after TAC (Supplementary Fig. 2a, b). Analysis of Gene-ontology terms (biological process) showed for example upregulation of proteasome and Foxo pathway genes, but downregulation of PI3K-Akt related genes, as indication of cachectic signaling (Supplementary Fig. 2c, d). Our aim was to identify previously unrecognized secreted factors that were dysregulated in skeletal muscle after TAC and that could affect the heart in an endocrine manner. We therefore filtered the primary RNAseq results for factors with a significant dysregulation, that were secreted (i.e. had a Gene Ontology (GO) term, cellular component, "extracellular region") and that were not highly expressed in the heart itself, since in that case, we assumed that the paracrine/autocrine influence of the respective factor would be more important than endocrine effects from skeletal muscle (Supplementary Fig. 2e). Our criteria were met by 48 mRNAs. One of these mRNA encoded for IGFBP5, a secreted IGF-1 binding protein that has previously been demonstrated to be downregulated in skeletal muscle of patients with heart failure[24]. For further analysis, we selected Musclin (encoded by the *Ostn* gene), which decreased by about 3.5-fold in the quadriceps muscle after chronic TAC (Supplementary Fig. 2e). By qPCR, we did not find significant expression of *Ostn* mRNA in mouse heart, kidney, fat, or liver tissue, but high expression in the quadriceps muscle, which was downregulated 12 weeks after TAC (Fig. 1s). We confirmed downregulation of Musclin protein after chronic TAC in the quadriceps muscle by Western blotting and immunostaining, as well as in mouse plasma (Fig. 1t–v). Analysis of *Ostn* mRNA levels in different wasting skeletal muscles (quadriceps, triceps and gastrocnemius muscles) revealed its continuous downregulation during disease progression (Fig. 1w).

**Patients with cardiac cachexia exert reduced skeletal muscle *OSTN* expression.** To investigate whether or not our findings might also be relevant in patients, we assessed *OSTN* mRNA

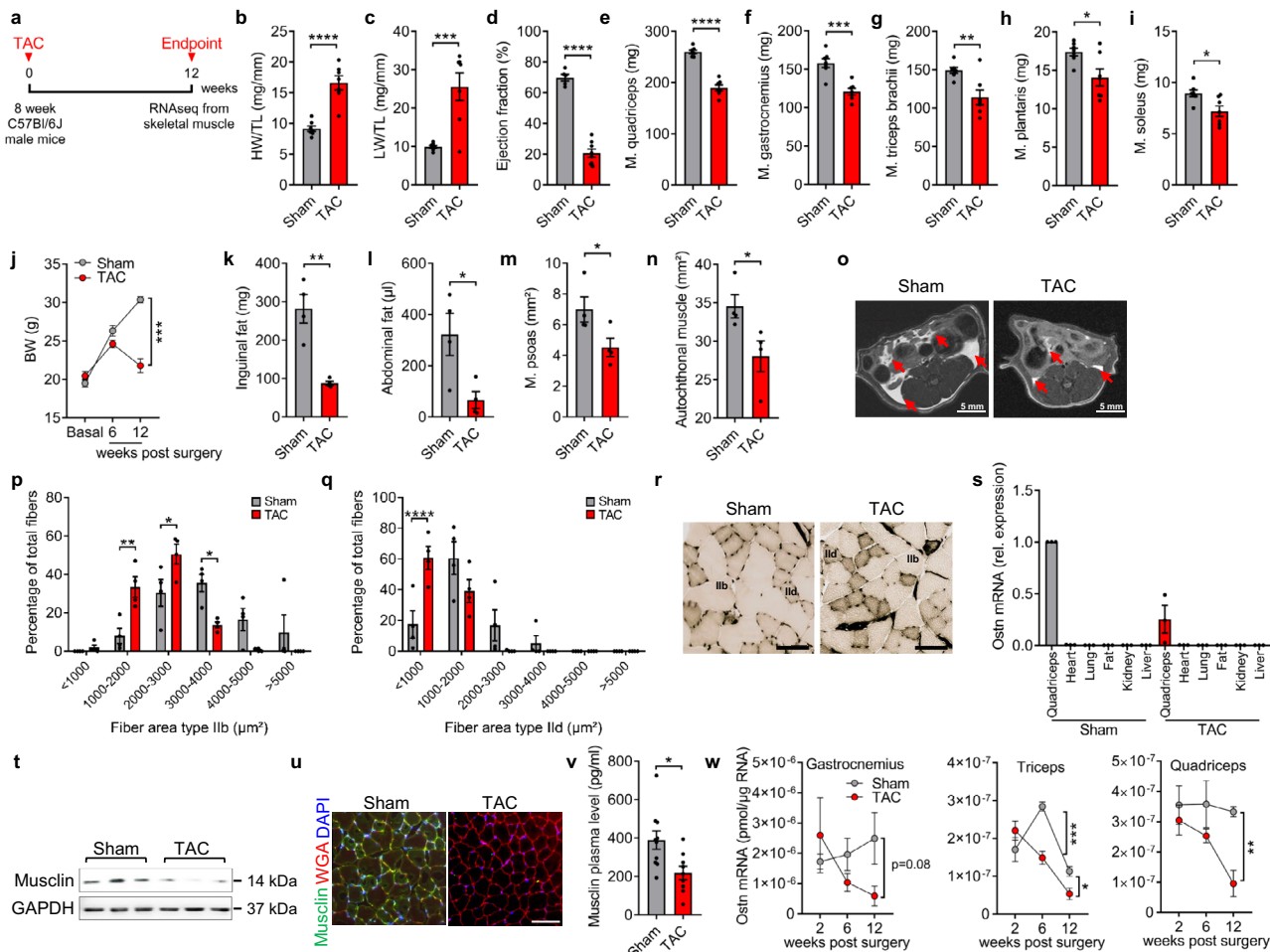

**Fig. 1 Musclin expression decreases in wasting skeletal muscles during advanced heart failure. a** Scheme depicting the experimental time line. Quantification of heart weight/tibia length (HW/TL) ratio, $n = 7$/group, ****$p < 0.0001$ (**b**), lung weight/tibia length (LW/TL) ratio, $n = 7$/group, ***$p = 0.001$ (**c**), left ventricular ejection fraction, $n = 5$/sham group and $n = 8$/TAC group, ****$p < 0.0001$ (**d**), quadriceps muscle weight, $n = 7$/group, ****$p < 0.0001$ (**e**), gastrocnemius muscle weight $n = 7$/group, ***$p = 0.0003$ (**f**), triceps brachii muscle weight, $n = 7$/group, **$p = 0.0051$ (**g**), plantaris muscle weight, $n = 7$/group, *$p = 0.0175$ (**h**), soleus muscle weight, $n = 7$/group, *$p = 0.0126$ (**i**), and course of body weight (BW) during the experiment until 12 weeks after TAC or sham surgery, $n = 4$/sham group and $n = 5$/TAC group, ***$p = 0.0001$ (**j**). Inguinal fat weight, **$p = 0.0021$ (**k**) in the group of mice that was analyzed by MRI to determine the amount of abdominal fat, *$p = 0.0273$ (**l**), psoas muscle, *$p = 0.0499$ (**m**) and autochthonal muscle, *$p = 0.0412$ (**n**) 12 week after sham or TAC surgery, $n = 4$/group. Example MRI cross-sections are shown in (**o**). The red arrows indicate abdominal fat. Scale bar: 5 mm. Cross-sectional fiber area type IIb, **$p = 0.0028$ for 1000–2000 μm²; *$p = 0.0254$ for 2000–3000 μm² and *$p = 0.0125$ for 3000–4000 μm² of muscle fiber area from sham vs. TAC group (**p**), and type IId, ****$p < 0.0001$ of fiber area <1000 μm² (**q**) of quadriceps muscles 12 weeks after sham or TAC surgery, $n = 4$/group, analyzed from microscopic pictures with ATPase staining (pH 4.2) as shown in (**r**). Scale bar: 300 μm. **s** Ostn (Musclin) mRNA levels in different organs 12 weeks after sham or TAC surgery, $n = 3$/group. **t** Immunoblot (the size of the proteins is indicated in kDa) and immunofluorescence staining (**u**) showing Musclin protein levels in the quadriceps muscle of sham and TAC mice. Scale bar: 100 μm. **v** Musclin plasma levels in mice 12 weeks after sham or TAC surgery, $n = 10$ for sham and $n = 9$ for TAC, *$p = 0.0115$. **w** Time course (2, 6 and 12 weeks after TAC or sham surgery), $n = 4$/group, showing the decline of Ostn mRNA levels in the gastrocnemius, triceps, ***$p = 0.0007$ 6 weeks and *$p = 0.0241$ 12 weeks, in quadriceps muscles, **$p = 0.002$ by 12 weeks after TAC compared to sham mice. Data are shown as mean ± standard error of the mean (SEM). *$p < 0.05$, **$p < 0.01$, ***$p < 0.001$, ****$p < 0.0001$ as determined by Kruskal–Wallis test with Dunn's multiple comparisons test (for **s**) or two-tailed Student's $t$ test (all other numerical data containing panels) or $p$ value determined by one-way ANOVA followed by the Holms–Sidak's multiple comparisons test (**p**, **q**). Source data are provided as a source data file.

levels in vastus lateralis muscle biopsy specimens from control individuals and from patients with sarcopenia or cachexia due to CHF. Sarcopenia was defined as reduced appendicular muscle mass (normalized to height), while cachexia was defined as non-edematous, non-intentional weight loss ≥ 5% over a period of 1 year according to the consensus definitions[25,26]. More detailed information on the investigated control individuals and patients is available in Table 1. As indicated in Fig. 2a, cachectic patients had markedly and significantly reduced skeletal muscle *OSTN* mRNA levels. Although *OSTN* mRNA expression levels were also

reduced in heart failure patients with sarcopenia, this did not reach statistical significance. We next investigated whether Musclin protein levels were also reduced in the serum of heart failure patients. To this end, we determined the Musclin concentration in serum of healthy individuals in comparison to patients suffering from heart failure due to advanced aortic stenosis. These patients were on average older than the healthy individuals, which is a limitation of this analysis (Table 2). Eight of these patients suffered from diabetes mellitus. In transthoracic echocardiography, they exerted a reduced aortic valve area and an

**Table 1 Clinical characteristics of the patients or control individuals, who received a muscle biopsy.**

|  | Control N = 9 | Sarcopenic N = 6 | Cachectic N = 5 |
|---|---|---|---|
| Age (years) | 56.08 ± 24.05 | 68.3 ± 5.5 | 55.6 ± 8.7 |
| Male/Female n (%) | 5/4 (56/44) | 5/1 (83/17) | 4/1 (80/20) |
| BMI (g/m²) | 24.47 ± 2.87 | 24.0 ± 4.3 | 29.1 ± 8.5 |
| Weight (kg) | 72.4 ± 12.79 | 70.5 ± 15.3 | 87.7 ± 23 |
| Weight loss 1 year (kg) | 0.33 ± 0.58 (n = 3) | −1 ± 2.9 | 5.1 ± 1.6* |
| Diabetes mellitus | 0 | 2 (28.6%) | 2 (40.0%) |
| Echocardiography |  |  |  |
| Ejection fraction (%) | 61.88 ± 3.13 | 32.5 ± 9.4**** | 32 ± 7.6**** |
| Clinical classification [% of all patients] |  |  |  |
| NYHA I | N/A | 2 (33.3%) | 1 (20%) |
| NYHA II | N/A | 2 (33.3%) | 3 (60%) |
| NYHA III | N/A | 1 (16.7%) | 1 (20%) |
| NYHA IV | N/A | 1 (16.7%) | 0 (0%) |
| Medication [% of all patients] |  |  |  |
| ACE inhibitor | 2 (40%) | 4 (66.7%) | 4 (80%) |
| Angiotensin receptor antagonist | None | 2 (33.3%) | 1 (20%) |
| β-receptor antagonist | None | 6 (100%) | 5 (100%) |
| Diuretic | None | 3 (50%) | 2 (40%) |
| Calcium channel antagonist | 1 (20%) | 1 (16.7%) | 1 (20 %) |

Data are displayed as mean ± SD, absolute number (n) or in % of all patients as indicated. BMI denotes body mass index. NYHA indicates New York Heart Association Class. ****$p < 0.0001$ vs. Control, *$p = 0.025$ vs. Control, determined by one-way ANOVA followed by the Holms–Sidak's multiple comparisons test.

failure patients (2.36 ng/ml) versus healthy individuals (3.38 ng/ml) (Fig. 2b). Because slightly increased Musclin serum levels were recently reported in patients with diabetes mellitus, we re-analyzed the data while excluding the diabetic patients[27]. Without diabetics, heart failure patients only had a mean serum Musclin concentration of 1.99 ng/ml (Fig. 2c).

**Adeno-associated virus 6 (AAV6) mediated overexpression of Musclin in skeletal muscle inhibits cardiac fibrosis and dysfunction during chronic pressure overload.** Since endogenous Musclin expression was reduced in wasting skeletal muscles during cardiac cachexia in mice and patients with heart failure, we assessed the functional consequence of Musclin overexpression in skeletal muscle as potential therapeutic approach in mice. Two days after sham or TAC surgery, we applied AAV6 Musclin (containing the muscle creatine kinase promoter to ensure skeletal muscle-specific expression) or AAV6 Control (overexpressing Luciferase protein) via injection into the right quadriceps muscle for long-term skeletal muscle-specific overexpression. The experiment was terminated 9 weeks after surgery (Fig. 3a). At this time point, qPCR and Western blotting revealed that overexpression of Musclin via AAV6 was robust and occurred specifically in the right quadriceps muscle, but not in the heart (Fig. 3b, c). In addition, Musclin protein levels were significantly increased in plasma of AAV6 Musclin versus control treated mice (Fig. 3d). Both groups of mice developed a comparable degree of cardiac hypertrophy in response to TAC, as shown by similarly enhanced HW/TL ratio, embryonic gene expression (decrease in *Myh6* and *Atp2a2*, increase in *Myh7* and *Acta1* mRNA) and cardiomyocyte size (Fig. 3e, Supplementary Fig. 3a–f). Assessment of systolic cardiac function by echocardiography revealed a marked reduction of the left ventricular ejection fraction in all mice 3, 6 and 9 weeks after TAC, but cardiac dysfunction was less pronounced in the AAV6 Musclin treated mice at all examined time points (Fig. 3f–h, Supplementary Table 1 showing additional echocardiographic parameters). A second cohort of mice was treated like shown in Fig. 3a, but was examined with earlier echocardiography as well as with a Doppler based quantification of the initial degree of pressure overload (Supplementary Fig. 4a). This analysis revealed a similar initial degree of pressure overload in AAV6 Control and AAV6 Musclin injected mice. In addition, echocardiography showed equal systolic cardiac function in both groups before TAC, as well as 1 week after TAC, but improved heart function developed in the AAV6 Musclin treated mice 3 weeks after TAC (Supplementary Fig. 4b–e). The delayed improvement of heart function in response to AAV6 Musclin application is in accordance with a delayed protein overexpression by the vector that typically takes around 5–7 days to start after injection[28,29]. Cardiac pressure-volume recordings by Millar catheterization showed increased left ventricular contractility (+dp/dt) and improved relaxation (−dp/dt) in AAV6 Musclin treated mice 9 weeks after TAC (Fig. 3i, j). Accordingly, intra-ventricular (maximal and mean) pressure was more increased after TAC in the left ventricle of AAV6 Musclin versus the AAV6 Control treated mice, likely due to higher myocardial contractility (Supplementary Fig. 3g, h). On the other hand, Musclin overexpression triggered mainly a reduction in peripheral systolic, and less in diastolic blood pressure, as it had been previously reported in Musclin overexpressing transgenic mice due to vascular relaxation (Supplementary Fig. 3i, j)[17]. Histological examinations revealed that overexpression of Musclin in the quadriceps muscle led to a marked reduction of myocardial fibrosis compared to AAV6 Control treated mice, while cardiac capillary density was similarly increased after TAC (Fig. 3k, l, Supplementary Fig. 3k). In line with the histological

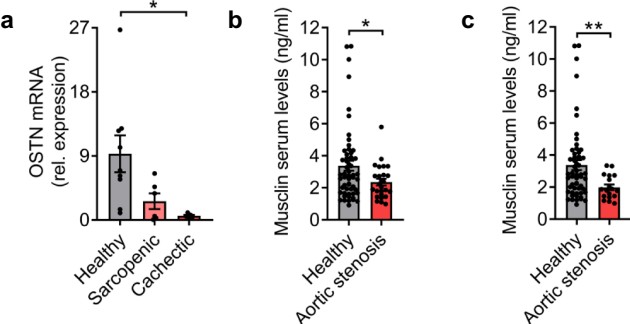

**Fig. 2 Muscle OSTN mRNA and Musclin serum levels are reduced in patients with heart failure. a** *OSTN* mRNA levels in vastus lateralis muscle biopsy samples from control individuals (n = 9) and from patients with sarcopenia (n = 6) or cachexia (n = 5), *$p = 0.0236$. **b** Concentrations of Musclin protein in the serum of healthy control individuals (n = 56) and of patients with severe aortic stenosis (n = 26), *$p = 0.0403$ (**c**) and of the same patients excluding the ones with diabetes mellitus (n = 18), **$p = 0.0044$. Data are shown as mean ± standard error of the mean (SEM). *$p < 0.05$, **$p < 0.01$ as determined by ANOVA followed by the Holms–Sidak's multiple comparisons test (**a**) and the two-tailed Mann–Whitney test (**b**, **c**). Source data are provided as a source data file.

increased mean transvalvular gradient, a reduced left ventricular ejection fraction, and an increased septum thickness as sign of left ventricular hypertrophy. 63% of the patients suffered from advanced heart failure symptoms, as they belonged to New York Heart Association (NYHA) classes III or IV (Table 2). The serum concentration of Musclin was significantly reduced in heart

**Table 2 Clinical characteristics of the patients (all male) with aortic stenosis.**

|  | Aortic stenosis $N = 26$ | Normal values | Healthy $N = 56$ |
|---|---|---|---|
| Age (years) | 79.4 ± 7.7**** |  | 60.5 ± 4.3 |
| BMI (g/m²) | 25.9 ± 3.1 | 18.5–25 |  |
| Diabetes mellitus (n) | 8 |  | 0 |
| Echocardiography |  |  |  |
| Ejection fraction (%) | 45.7 ± 15.9 (n = 17) | >55 |  |
| Septum thickness (mm) | 14.5 ± 0.5 (n = 25) | ≤11 |  |
| Left ventricular enddiastolic diameter (mm) | 48.9 ± 6 (n = 25) | 42–58 |  |
| Aortic valve gradient, mean (mmHg) | 37.4 ± 16.1 | ≤20 |  |
| Aortic valve area (cm²) | 0.7 ± 0.18 | >1.5 |  |
| Serum |  |  |  |
| Musclin (ng/ml) | 2.36 ± 1* |  | 3.38 ± 2.3 |
| Creatinine (µmol/l) | 124.5 ± 58 | 59–104 |  |
| Clinical classification [% of all patients] |  |  |  |
| NYHA I | 15.4 |  |  |
| NYHA II | 23.1 |  |  |
| NYHA III | 46.2 |  |  |
| NYHA IV | 15.4 |  |  |
| Medication [% of all patients] |  |  |  |
| ACE inhibitor | 50 |  |  |
| Angiotensin receptor antagonist | 19.2 |  |  |
| β-receptor antagonist | 69.2 |  |  |
| Diuretic | 69.2 |  |  |
| Calcium channel antagonist | 38.5 |  |  |

The mean age and Musclin levels of the healthy individuals (also all male) are shown in addition to normal values.
Data are displayed as mean ± SD or in % of all patients as indicated. BMI denotes body mass index. NYHA indicates New York Heart Association Class. *$p = 0.0403$ vs. Healthy (determined by two-tailed Mann–Whitney test) and ****$p < 0.0001$ (determined by two-tailed Student's $t$ test).

examinations, we detected increased expression of pro-fibrotic genes (Postn, Col1a1, Col3a1, Fn1) in the myocardium of AAV6 Control treated mice after TAC, which were significantly decreased by AAV6 Musclin treatment (Fig. 3m–p). A third cohort of mice was treated as shown in Fig. 3a, but was taken until 12 weeks after surgery (Supplementary Fig. 5a). Musclin overexpression was still very effective in preventing pulmonary congestion, and also improved systolic heart function at this later stage (although only a trend was observed), but hypertrophy was still not influenced (Supplementary Fig. 5b–e). Interestingly, the decrease in body weight and skeletal muscle weight loss was partially prevented by Musclin overexpression (Supplementary Fig. 5f–h). Since it was proposed that Musclin acts by increasing the levels of natriuretic peptides, we determined the cardiac mRNA expression level of these peptides and their receptors. We found that Nppa (ANP) and Nppb (BNP) mRNA expression was significantly increased after TAC, but was not different between AAV6 Control and AAV6 Musclin treated mice (Supplementary Fig. 3l, m). The mRNA expression of Nppc (CNP) as well as that of Npr2-3 were not significantly changed between groups (Supplementary Fig. 3n–q). Npr1 mRNA expression was significantly downregulated in response to TAC in AAV Control treated mice. In line with reduced degradation of natriuretic peptides in response to Musclin overexpression, we detected a trend towards increased ANP levels and significantly increased mature CNP levels in plasma of AAV6 Musclin treated mice (Supplementary Fig. 3r, Fig. 3q). Collectively, overexpression of Musclin in skeletal muscle ameliorated cardiac dysfunction and myocardial fibrosis during long-term pressure overload.

**Skeletal muscle-specific disruption of Musclin during short-term TAC mimics the cardiac phenotype in response to long-term pressure overload.** To test whether skeletal muscle Musclin deficiency would be sufficient to aggravate cardiac dysfunction during cardiac overload in vivo, we generated mice with skeletal muscle specific, inducible deletion of Musclin using the Cre/LoxP

system. LoxP sites were introduced flanking exon 3 of the Musclin encoding Ostn gene (to generate Ostn^fl/fl mice), since elimination of this exon results in deletion of large parts of the coding sequence with a subsequent frame shift (Fig. 4a). Ostn^fl/fl mice were crossed with HSA-rtTA/TRE-Cre double transgenic mice that express Cre recombinase only in skeletal muscle and only following doxycycline (Dox) treatment (Fig. 4b). The first transgene uses the human skeletal actin (HSA) promoter to drive expression of the reverse tetracycline-controlled transactivator (rtTA). The second transgene uses a tetracycline responsive promoter to drive the expression of the Cre recombinase. Resulting Ostn^fl/fl (HSA/TRE-)Cre (referred to as Musclin knockout (KO) hereafter) and littermate Ostn^fl/fl control mice (WT) were treated for one week with Dox and were then exposed to pressure overload by TAC for 2 weeks, when endogenous skeletal muscle Musclin levels are still not changed versus sham surgery (Figs. 4c and 1w). Musclin mRNA and protein levels were markedly reduced in the quadriceps muscle of Musclin KO versus control mice as demonstrated by qPCR, immunoblotting and immunofluorescence (Fig. 4d–f). These changes were accompanied by a significant, but only partial reduction of Musclin plasma levels (Fig. 4g). A comparison between different tissues showed strongly diminished Musclin mRNA expression in the quadriceps muscle in KO mice, but no changes in bone and brain between both genotypes, which might explain the incomplete reduction in Musclin plasma levels in these mice (Fig. 4h). Musclin KO and WT mice similarly developed cardiac hypertrophy (increased HW/TL ratio, cardiomyocyte area and embryonic gene expression) and increased capillary density following TAC compared to sham surgery (Fig. 4i and Supplementary Fig. 6a–f). Musclin KO mice, however, showed enhanced left ventricular dysfunction in echocardiography and a significantly reduced systolic contractility (+dp/dt) as well as a reduced diastolic relaxation (−dp/dt) during Millar catheter examination after TAC (Fig. 4j–l, Supplementary Table 2). Systolic, but not diastolic blood pressure was slightly increased in Musclin KO mice after TAC (Supplementary Fig. 6g, h). Histological analyses revealed increased myocardial fibrosis in Musclin KO mice in

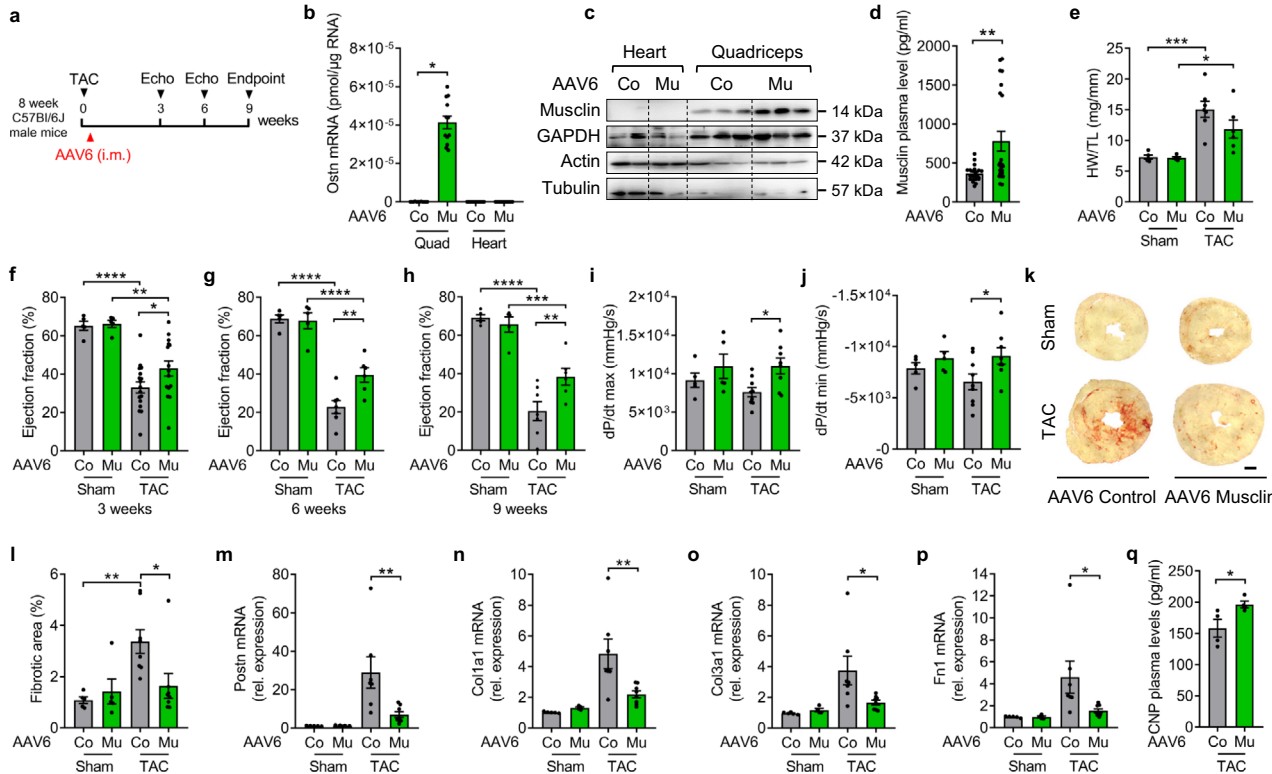

**Fig. 3 AAV6-mediated overexpression of Musclin in skeletal muscle attenuates left ventricular dysfunction and myocardial fibrosis during long-term pressure overload. a** Scheme depicting the experimental time line. **b** mRNA level of *Ostn* (Musclin) 9 weeks after TAC and intramuscular application of AAV6 control (Co) or AAV6 Musclin (Mu) vector (n = 12/group), *p = 0.036 (**c**) Immunoblot for Musclin 9 weeks after TAC and intramuscular application of AAV6 Co or AAV6 Mu. GAPDH, Actin and Tubulin are loading controls. The size of the proteins is indicated in kDa. **d** Musclin plasma levels in AAV6 Co (n = 21) or AAV6 Mu (n = 22) treated mice, **p = 0.0029. **e** Heart weight/tibia length (HW/TL) ratio in AAV6 Co (sham n = 5, TAC n = 7) or AAV6 Mu (sham n = 5, TAC n = 6) treated mice 9 weeks after sham or TAC surgery, ***p = 0.0004, *p = 0.0316. **f–h** Echocardiographic ejection fraction 3 weeks (all sham n = 5/group, TAC AAV6 Co n = 17, TAC AAV6 Mu n = 15), ****p < 0.0001, **p = 00015 and *p = 0.0293 (**f**), and 6 weeks after surgery, ****p < 0.0001 in AAV6 Control and AAV6 Musclin groups for sham vs. TAC surgery, **p = 0.0021 (**g**), and 9 weeks after surgery, ****p < 0.0001, ***p = 0.0006 and **p = 0.0051 (**h**). Cardiac systolic contractility (dp/dt max, **i**), *p = 0.0122 and diastolic relaxation (dp/dt min, **j**), *p = 0.0164 determined by left ventricular catheterization in the indicated mice (all sham n = 5/group, TAC AAV6 Co n = 9, AAV6 Mu n = 8) 9 weeks after sham or TAC surgery. Representative Sirius red-stained heart sections (**k**) and quantified myocardial fibrotic area (**l**) and of mice treated as shown (all sham n = 5/group, all TAC n = 8/group, 9 weeks after surgery), **p = 0.0093 and *p = 0.0206. Scale bar: 500 μm. **m–p** qPCR analysis of the indicated fibrosis genes 9 weeks after sham or TAC surgery (sham AAV6 Co n = 5, sham AAV6 Mu n = 4, TAC AAV6 Co n = 7, TAC AAV6 Mu n = 8), **p = 0.0064 (**m**), **p = 0.0081 (**n**), *p = 0.0248 (**o**), *p = 0.0369 (**p**). **q** Mature CNP plasma levels in AAV6 Co or AAV6 Mu treated mice 3 weeks after TAC surgery (n = 4/group), *p = 0.0462. Data are shown as mean ± standard error of the mean (SEM). *p < 0.05, **p < 0.01, ***p < 0.001, ****p < 0.0001 as determined by two-tailed Student's *t* test (**d**, **q**) or by one-way ANOVA followed by the Holms–Sidak's multiple comparisons test (other bar graphs). Source data are provided as a source data file.

response to TAC compared to WT mice (Fig. 4m, n), which was accompanied by upregulation of pro-fibrotic genes like *Postn*, *Col1a1*, *Col3a1* and *Fn1* (Supplementary Fig. 6i–l). Plasma levels of mature CNP were lower in KO versus WT mice after TAC (Fig. 4o), suggesting its enhanced degradation when Musclin is deleted in skeletal muscles. At this early time point, body weight and skeletal muscle weights were not significantly changed between groups (Supplementary Fig. 6m–o). Together, these results suggested that Musclin deficiency in skeletal muscle critically accelerates heart failure progression following short-term pressure overload and mimics the cardiac phenotype observed in WT mice following long-term TAC (when endogenous Musclin becomes downregulated).

**Musclin augments cardiomyocyte contractility via CNP/NPR-B signaling.** Because overexpression of Musclin in skeletal muscle of WT mice after TAC led to improved cardiac contractility, we hypothesized that these characteristic effects were attributable to an

elevated local CNP concentration through blockade of its NPR-C dependent degradation by Musclin binding to NPR-C and thus through enhanced CNP/NPR-B signaling (Fig. 5a). In this case, Musclin would compete with CNP for its binding to NPR-C. Our hypothesis was further based on the fact that CNP/NPR-B preferentially exerts positive inotropic and lusitropic effects, while ANP/NPR1 mediated pathways show anti-hypertrophic and anti-hypertensive, but no positive inotropic properties[30,31]. To address whether Musclin in principal can compete with CNP for its binding to NPR-C, we designed an ELISA-based assay, in which we coated plastic wells with NPR-C and applied Musclin together with increasing concentrations of CNP. Musclin that bound to NPR-C in the well was subsequently detected by an ELISA method (Fig. 5b). As shown in Fig. 5c, increasing concentrations of CNP were able to prevent binding of Musclin to NPR-C in the well and thereby led to Musclin displacement, indicating that indeed CNP and Musclin compete for binding NPR-C. To evaluate whether Musclin could lead to increased local CNP concentrations by this mechanism, we transfected HEK293 cells with an NPR-C-GFP

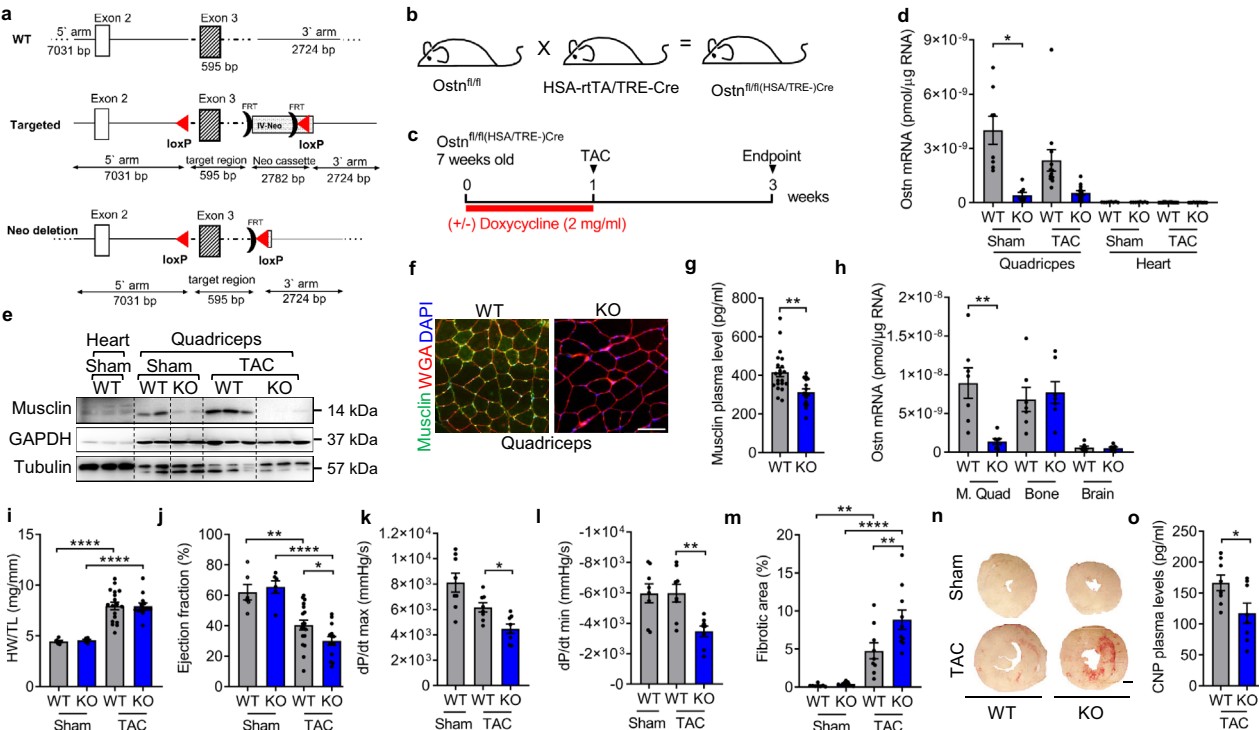

**Fig. 4 Adult induced skeletal muscle-specific Musclin deficient mice show exaggerated LV-dysfunction and myocardial fibrosis after short-term TAC.**
**a** Scheme depicting the Musclin (*Ostn*) knockout (KO) targeting strategy. The exon 3 of the *Ostn* gene is floxed and **b** deleted through the crossing with double transgenic mice that express Cre recombinase selectively in skeletal muscle and only following doxycycline treatment. **c** Scheme depicting the experimental time line. **d** qPCR analysis of *Ostn* mRNA in the quadriceps muscle and the heart 2 weeks after TAC or sham surgery in KO vs. littermate control (WT) mice (all sham n = 8/group, all TAC n = 12/group), *p = 0.0359. **e** Immunoblot for Musclin in the quadriceps muscle and the heart 2 weeks after TAC or sham surgery in WT vs. KO mice. GAPDH and Tubulin are loading controls. The size of the proteins is indicated in kDa. **f** Immunofluorescence staining for Musclin in the quadriceps muscle of WT and KO mice. Scale bar: 100 μm. **g** Musclin plasma levels of WT (*n* = 21) and KO mice (*n* = 16), **p = 0.0014. **h** *Ostn* mRNA levels in the indicated organs of the mice as shown (n = 7/group), **p = 0.0027. Quantification of the heart weight/tibia length ratio (HW/TL, ****p < 0.0001) (**i**), echocardiographic ejection fraction, **p = 0.0018 and ****p < 0.0001, *p = 0.0445 (**j**), systolic contractility (**k**, by LV-catheter, *p = 0.0334) and diastolic relaxation (**l**, by LV-catheter, **p = 0.0094) in mice treated as indicated in (**c**), (for HW/TL ratio and for echocardiography: all sham n = 6/group, TAC WT n = 18, TAC KO n = 14; for LV-catheter: n = 8/group). **m, n** Myocardial fibrotic area with representative Sirius red-stained myocardial sections of WT and KO mice treated as indicated (all sham n = 8/group, TAC WT n = 9, TAC KO n = 10, **p = 0.0033 in WT and ****p < 0.0001 in KO mice after sham vs. TAC surgery, **p = 0.0033 in WT vs. KO mice after TAC). Scale bar: 500 μm. **o** Mature CNP peptide plasma levels in the indicated mice (n = 8/group, *p = 0.0312). Data are shown as mean ± standard error of the mean (SEM). *p < 0.05, **p < 0.01, ****p < 0.0001 determined by two-tailed Student's *t* test (**g**, **h**, **o**) or one-way ANOVA followed by the Holms–Sidak's multiple comparisons test (other bar graphs). Source data are provided as a source data file.

fusion construct (or left them untransfected) (Fig. 5d). We detected a trend towards reduced CNP levels in the supernatant of NPR-C-transfected HEK293 cells versus untransfected cells in the absence of Musclin, but Musclin significantly enhanced CNP levels in the supernatant of transfected cells, implying that indeed Musclin blocks NPR-C dependent CNP degradation (Fig. 5e). In addition, we tested the inotropic response of isolated cardiomyocytes to CNP and subsequent Musclin co-stimulation. Indeed, adult murine cardiomyocytes isolated from WT mice exhibited improved cardiomyocyte contractility following incremental concentrations of CNP (10, 100 nM), which was augmented by subsequent co-stimulation with 10 nM Musclin (Fig. 5f, g). Notably, the improved contractile function of cardiomyocytes following CNP stimulation was accompanied by increased cytosolic free Ca²⁺ concentration in ratiometric recordings of Fura-2-loaded cultured cardiomyocytes, which was again augmented by 10 nM Musclin co-stimulation (Fig. 5h, i). Musclin treatment alone exhibited no effect on cardiomyocyte contractility (Supplementary Fig. 7a, b). In addition, we verified that increased inotropy by Musclin was not mediated by ANP signaling, since no pro-contractile effects were detectable when isolated cardiomyocytes were stimulated with 100 nM ANP and subsequent 10 nM Musclin co-stimulation

(Supplementary Fig. 7c, d). This was despite the fact that ANP increased intracellular cGMP levels in WT cardiomyocytes, which was enhanced by co-stimulation with Musclin, whereas neither ANP nor Musclin promoted cGMP levels in NPR-A deficient cardiomyocytes (Supplemental Fig. 7e, f).

We next wanted to address how Musclin enhances cardiomyocyte contractility in the presence of CNP. To this end, we analyzed cardiomyocyte contractility in NPR-A or NPR-B deficient cardiomyocytes in comparison to WT cardiomyocytes. Stimulation with CNP increased cardiomyocyte contractility, which was enhanced by co-stimulation with Musclin in WT as well as in NPR-A deficient cardiomyocytes. In contrast, NPR-B deficient cardiomyocytes did not exert an increased contractility to CNP or the combined stimulation with CNP and Musclin. These data indicate that augmented CNP effects by Musclin are mediated mainly via NPR-B-dependent signaling (Fig. 5j–l).

**Musclin triggers enhanced cardiomyocyte cAMP generation through cGMP-mediated inhibition of PDE3.** The second messenger cGMP transduces CNP/NPR-B signaling either through inhibition of phosphodiesterase 3 (PDE3) thereby increasing cAMP levels via a positive cGMP-to-cAMP crosstalk

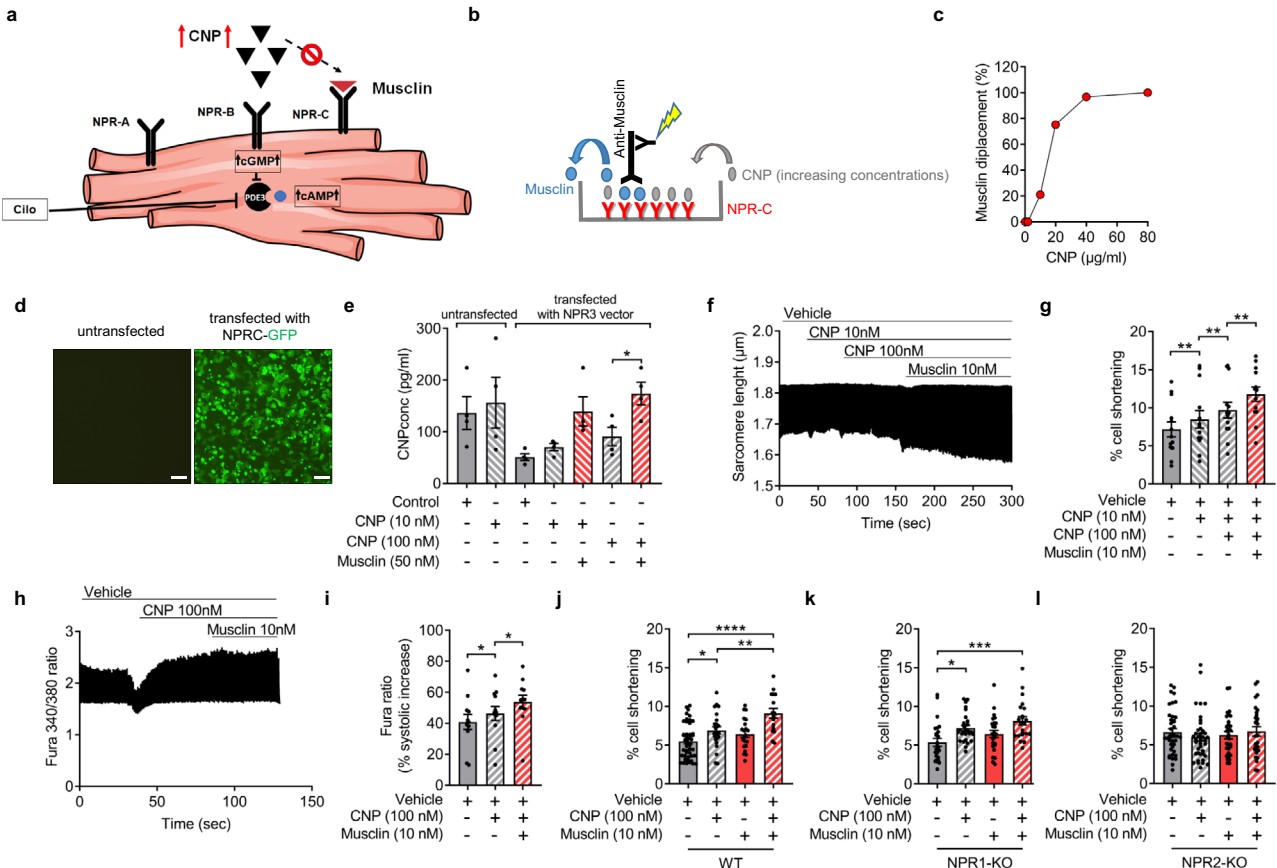

**Fig. 5 Musclin augments cardiomyocyte contractility by enhancement of CNP/NPR-B signaling. a** Scheme depicting the proposed mechanism of augmented cardiomyocyte contractility by Musclin. Scheme (**b**) and results (**c**) of the NPR-C/Musclin/CNP competition assay. Increasing levels of recombinant CNP led to Musclin displacement from NPR-C ($n = 2$ per CNP concentration, mean values are shown). **d** Representative fluorescence images from Hek293 cells with and without transfection of the NPR-C-GFP construct. Scale bar 100 μm. **e** CNP concentrations (determined by ELISA) in the supernatant of transfected and untransfected Hek293 cells (as shown in **d**) 1 h after treatment as indicated ($n = 4$/condition, *$p = 0.0104$). **f** Representatives traces of sarcomere length and **g** quantification of cell shortening (from traces as shown in **f**) in isolated wild-type mouse cardiomyocytes treated with recombinant CNP or Musclin as indicated ($n = 13$ cardiomyocytes/group, **$p = 0.0016$ for Vehicle vs. CNP (10 nM), **$p = 0.0012$ for CNP (10 nM) vs. CNP (100 nM) and **$p = 0.0026$ for CNP (100 nM) vs. CNP (100 nM) plus Musclin (10 nM)). **h** Representative Fura-2 Ca$^{2+}$ traces and quantitative analysis (**i**) in cardiomyocytes treated as described for (**f**) ($n = 12$ cells/group, *$p = 0.0168$ for Vehicle vs. CNP and *$p = 0.0169$ for cells stimulated with CNP vs. Musclin). **j–l** Cell shortening in cardiomyocytes treated as indicated from wild-typ (WT) (vehicle cardiomyocytes $n = 46$, after CNP stimulation $n = 24$, treated with Musclin $n = 21$, cardiomyocytes treated with CNP and Musclin $n = 16$ cells), ****$p < 0.0001$, *$p = 0.0278$ and **$p = 0.0091$ (**j**), cardiomyocyte-specific Npr1 knockout (vehicle-treated cardiomyocytes $n = 25$, after CNP stimulation $n = 27$, cardiomyocytes treated with Musclin $n = 25$, cardiomyocytes treated with CNP and Musclin $n = 22$, *$p = 0.01$ and ***$p = 0.0003$ (**k**), and global Npr2 knockout mice (vehicle-treated cardiomyocytes $n = 40$, after CNP stimulation $n = 39$, treated with Musclin $n = 31$, cardiomyocytes treated with CNP and Musclin $n = 28$ cells) (**l**). Data in bar graphs are shown as mean ± standard error of the mean (SEM). *$p < 0.05$, **$p < 0.01$, ***$p < 0.001$, ****$p < 0.0001$ determined by Kruskal–Wallis with Dunn's multiple comparisons test (**e**) or by one-way ANOVA followed by the Holms–Sidak's multiple comparisons test (all other panels). Source data are provided as a source data file.

or through stimulation of the cGMP dependent protein kinase G (PKG). To analyze whether the augmented inotropic response is mediated through the cGMP-inhibited cAMP-degrading PDE3, we used a FRET-based approach to visualize cGMP and cAMP dynamics in real time in cardiomyocytes from either cGES-DE5 transgenic (for cGMP measurements) or from Epac2-camps (for cAMP measurements) transgenic mice[32,33]. Indeed, cGMP concentrations increased in cardiomyocytes in response to 100 nM CNP, which were augmented under co-stimulation with 100 nM Musclin (Fig. 6a, b). Next, cAMP concentrations during exposure to 100 nM CNP were measured as percent of the effect induced by pre-stimulation with isoproterenol, which activates the sensor (FRET, in % of maximum). Fig. 6c depicts representative tracings showing that CNP and even more the co-stimulation with 100 nM Musclin elevated cAMP levels (Fig. 6d). We then addressed the relevance of NPR-B for cardiomyocyte cGMP and

cAMP levels (measured with specific ELISA assays) during CNP and Musclin stimulation. As depicted in Fig. 6e–h, CNP and even more the combined CNP and Musclin stimulation enhanced cGMP and also cAMP levels in WT, but not in *Npr2* KO cardiomyocytes, indicating that positive cGMP-to-cAMP crosstalk due to CNP and Musclin is mediated mainly via NPR-B.With regard to signaling downstream of NPR-B, inhibition of PDE3 with 10 μM cilostamide in WT cardiomyocytes prevented the increase in cAMP levels, as well as the enhancement of contractility to Musclin and CNP stimulation (Fig. 6i–l). In contrast, PKG as the other main target of cGMP might not participate in this inotropic effect, as application of the PKG inhibitor DT3 did not alter the pro-contractile effect of combined CNP/Musclin stimulation (Supplementary Fig. 7g, h). Therefore, treatment with cilostamide mimicked the improved cardiomyocyte contractility by Musclin following CNP stimulation and prevented its further

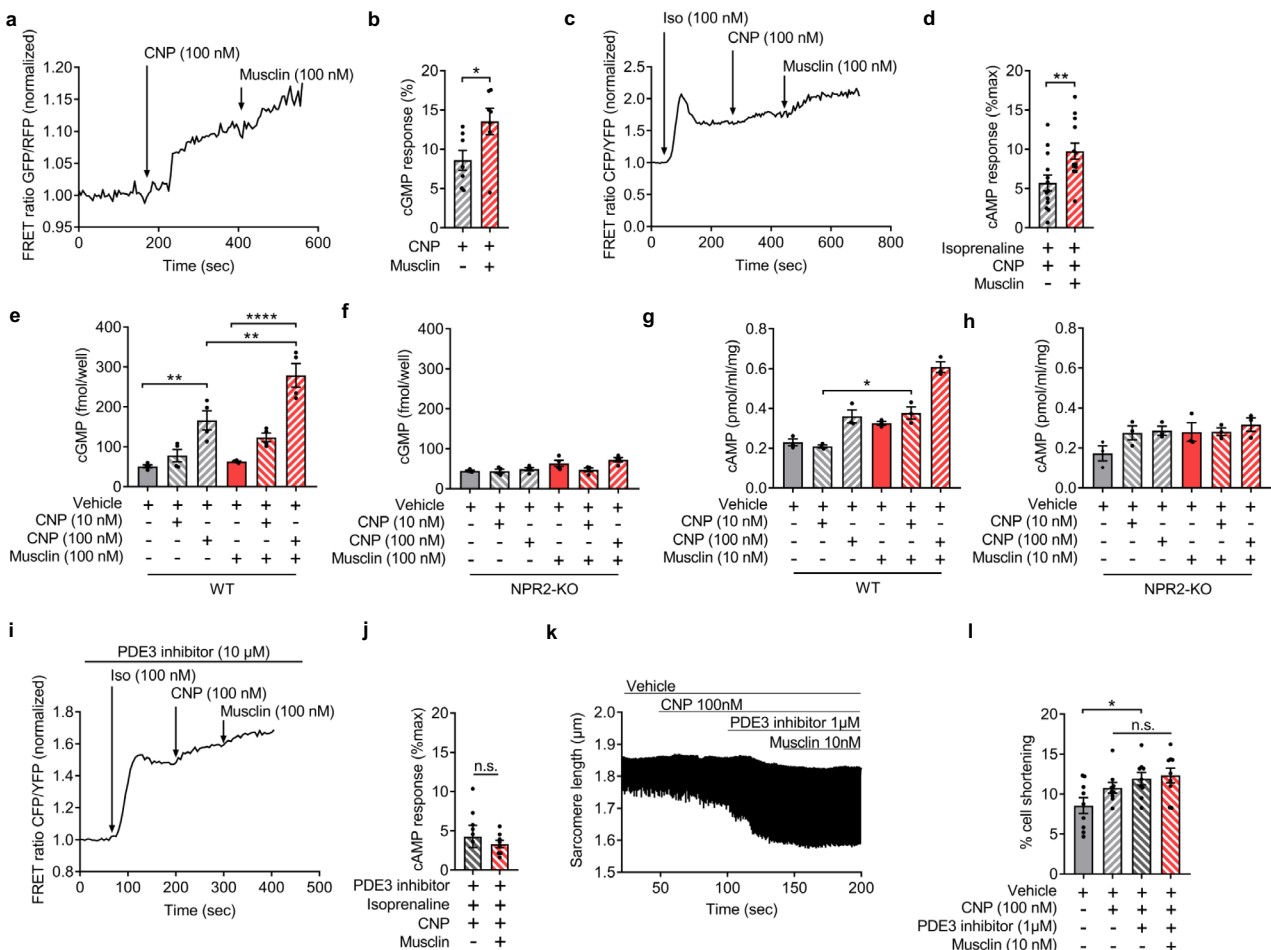

**Fig. 6 Musclin increases cGMP and cAMP levels and cardiomyocyte contractility through NPR-B-dependent inhibition of PDE3.** Cardiomyocytes were isolated from either cGES-DE5 transgenic (for cGMP measurements) or from Epac2-camps (for cAMP measurements) transgenic mice. **a, b** Fluorescence resonance energy transfer (FRET)-based measurements of cGMP in single cultured cardiomyocytes ($n = 7$ cells). The stimulation with CNP and Musclin was conducted as indicated. Representative traces (**a**) and a quantitative analysis (**b**) are shown, *$p = 0.0342$. **c, d** FRET-based measurements of cAMP in single cardiomyocytes treated as indicated ($n = 13$ cells). The cells were stimulated with Isoproterenol (Iso), CNP and subsequently Musclin as indicated. The graph illustrates the FRET-responses to CNP in % of the maximal Iso effect. Representative traces (**c**) and a quantitative analysis (**d**) are shown, **$p = 0.0019$. **e–h** ELISA-based measurements of cGMP and cAMP in cultured cardiomyocytes under the indicated conditions from wild-typ (WT) (**e, g**), **$p = 0.0012$ for vehicle-treated cells vs. stimulated with CNP (100 nM), ****$p < 0.0001$ for cells stimulated with Musclin vs. with CNP (100 nM) and Musclin, **$p = 0.0013$ for cells treated with CNP (100 nM) vs. cells stimulated with Musclin and CNP (100 nM) (**e**), *$p = 0.034$ (**g**) and global Npr2 knockout mice (**f, h**) (for cGMP measurement $n = 4$/condition and for cAMP $n = 3$/condition). **i, j** FRET-based measurements of cAMP in single WT cardiomyocytes treated as indicated. Representative traces (**i**) and a quantitative analysis (**j**) are shown ($n = 6$ cells without Musclin treatment and $n = 8$ cells with Musclin). **k** Representatives traces of sarcomere length and **l** quantification of cell shortening (from traces as shown in **k**) in isolated wild-type mouse cardiomyocytes treated as indicated ($n = 9$ cardiomyocytes per group, *$p = 0.017$). Cilostamide was used as PDE3 inhibitor. Data in bar graphs are shown as mean ± standard error of the mean (SEM). *$p < 0.05$, **$p < 0.01$, ****$p < 0.0001$ determined by two-tailed Student's $t$ test (**b, d, j**), one-way ANOVA followed by the Holms–Sidak's multiple comparisons test (**e, f, l**) or by Kruskal–Wallis with Dunn's multiple comparisons test (**g, h**). n.s. denotes "not significant". Source data are provided as a source data file.

augmentation during subsequent co-stimulation with Musclin, indicating that the increase in cardiomyocyte contraction is mediated through PDE3 inhibition.

**Skeletal muscle Musclin regulates cardiomyocyte cGMP and cAMP levels and the phosphorylation of phospholamban.** As the next step, we interrogated cGMP and cAMP levels in cardiomyocytes of mice with skeletal muscle Musclin deletion or its AAV-mediated overexpression. We found that cGMP and cAMP levels increased during TAC in WT mice, but were both markedly reduced in Musclin KO mice (Fig. 7a, b). Mice with skeletal muscle Musclin overexpression, in turn, displayed upregulated cardiomyocyte cGMP and cAMP concentrations after TAC

(Fig. 7c, d). To determine the downstream effects of Musclin triggered cAMP increases, we interrogated the phosphorylation of phospholamban (PLB) at its serine residue 16 (a known cAMP/protein kinase A (PKA) target) in mice with decreased or increased Musclin levels. Enhanced serine 16 PLB phosphorylation leads to increased cardiomyocyte contraction and relaxation[34]. As depicted in Fig. 7e, f, skeletal muscle-specific KO of Musclin reduced the phosphorylation of PLB in cardiomyocytes and hearts 3 and 14 days after TAC, respectively. Musclin overexpression, in turn, increased serine 16 phosphorylation of PLB (Fig. 7g). Serca2a protein levels, in contrast, were not different between the different groups (Fig. 7e, g). Hence, our data suggest that Musclin improves cardiomyocyte contractility

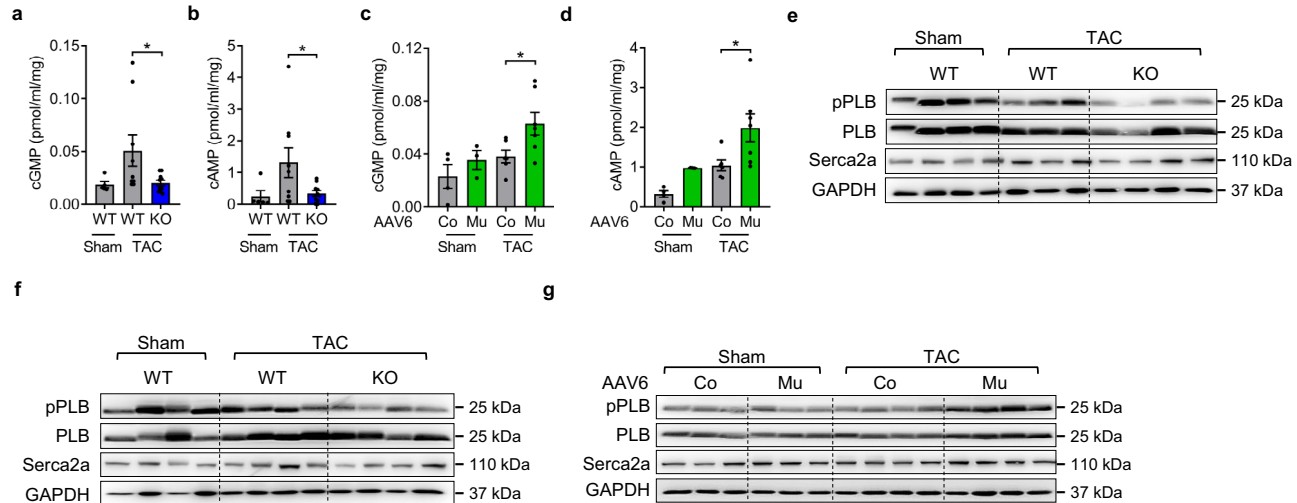

**Fig. 7 Skeletal muscle Musclin regulates cardiac cGMP and cAMP levels and phospholamban phosphorylation at the PKA phosphorylation site (Serine 16).** Cardiomyocyte cGMP (**a**, *$p = 0.0268$), (**c**, *$p = 0.048$) and cAMP (**b**, *$p = 0.029$), (**d**, *$p = 0.025$) levels (determined by ELISA) after sham and TAC surgery in control (WT) and Musclin knockout (KO) mice after TAC, as well as in Sham and TAC operated mice treated either with AAV6 Control (Co) or AAV6 Musclin (Mu) (WT sham $n = 5$, WT TAC $n = 9$, KO TAC $n = 10$, sham AAV6 Co $n = 4$, sham AAV6 Mu $n = 3$, TAC AAV6 Co $n = 6$, TAC AAV6 Mu $n = 7$). Immunoblots for the indicated proteins (GAPDH as loading control) from cardiac protein lysate 3 days (**e**), cardiomyocyte protein lysate 14 days (**f**) or cardiac protein lysate 9 weeks (**g**) after TAC or sham surgery in WT or KO mice or in mice treated with AAV6 Co or AAV6 Mu as shown. The size of the proteins is indicated in kDa. Data in bar graphs are shown as mean ± standard error of the mean (SEM). *$p < 0.05$, determined by one-way ANOVA followed by the Holms–Sidak's multiple comparisons test. Source data are provided as a source data file.

through PDE3 inhibition by augmentation of CNP/NPR-B/cGMP signaling, stimulation of a positive cGMP-to-cAMP crosstalk and activation of PKA signaling in cardiomyocytes.

**Musclin inhibits cardiac fibroblast proliferation and migration through activation of PKG.** We next assessed the impact of Musclin on isolated primary cardiac fibroblasts, to analyze whether it acts on these cells to explain the changes in TAC triggered myocardial fibrosis in response to skeletal muscle overexpression or ablation of Musclin. As shown in Fig. 8a, b, CNP (100 nM) and Musclin (50 nM) inhibited cardiac fibroblast proliferation and attenuated their capacity to migrate after a scratch injury, although almost no additive effect of CNP/ Musclin co-stimulation was observed. The latter was in contrast to what we found in isolated cardiomyocytes and could be due to the fact that fibroblasts produce CNP on their own, and therefore the effect might be saturated and could not be further enhanced by external CNP. The inhibitory effects of Musclin and CNP on fibroblast proliferation and migration were abolished by downregulation of NPR-B via siRNA, but were essentially unchanged by siRNA mediated downregulation of NPR-A, or by treatment with control siRNA (Fig. 8c–h, Supplementary Fig. 8a, b). This indicated that sufficient expression of NPR-B, but not of NPR-A was necessary for Musclin and/or CNP mediated inhibition of fibroblast activity. Downregulation of NPR-C also did not ablate and even partially enhanced the effects of CNP and Musclin, especially with regard to cell migration (Fig. 8c–h, Supplementary Fig. 8c). In this case, partial downregulation of NPR-C by siRNA had a similar effect as Musclin, which blocks NPR-C, and therefore both maneuvers act in the same direction to antagonize NPR-C. Although this gives a hint on the role of NPR-C for Musclin dependent effects, a complete ablation/KO of NPR-C would have given a better clue in this regard. Because PKG is a known negative regulator of fibroblast activation[35], we investigated the impact of pharmacological PKG inhibition. The PKG inhibitor DT3 (200 nM) eliminated the Musclin effects on fibroblast proliferation and migration, suggesting that it acts via an enhancement of CNP/NPR-B/PKG signaling (Fig. 8i, j).

Western blot analyses (Fig. 8k) revealed that Musclin entailed a decreased activation of the p38 MAP kinase (MAPK), which is known to be responsible for the activation of cardiac fibroblasts and which can be inhibited by PKG[36–38]. The ERK MAPK was much less affected by Musclin. Addition of the PKG inhibitor (DT3) abolished the inhibitory effects of Musclin on p38 (Fig. 8k). Together, these experiments indicated that Musclin acts on cardiac fibroblasts by enhancing CNP/NPR-B/PKG signaling, which counteracts fibroblast activation by interfering with p38 MAPKs (Supplementary Fig 8d).

**Discussion**
In this study, we identified reduced levels of Musclin in wasting skeletal muscle as driver of heart failure progression. Musclin is expressed in skeletal muscle and in bone, where it was shown to mediate insulin-dependent glucose metabolism and endochondral growth, respectively[15,16,39–43]. Musclin has a strong homology to the natriuretic peptide family and is an endogenous ligand of the natriuretic peptide clearance receptor (NPR-C)[44]. Although Musclin can also act locally in skeletal muscle to enhance physical endurance by promoting mitochondrial biogenesis[12,45,13], it is also released to plasma, suggesting a systemic impact on the natriuretic peptide system. The distinct downregulation of Musclin during long-term pressure overload in skeletal muscles suggested a specific impact of this myokine on heart failure progression via inter-tissue crosstalk. Indeed, therapeutic overexpression of Musclin in skeletal muscle via AAV6 ameliorated left ventricular dysfunction and myocardial fibrosis after long-term TAC in mice. Moreover, by generation of a skeletal muscle specific and inducible Musclin KO mouse, we revealed that Musclin deficiency in skeletal muscle triggers exaggerated maladaptive cardiac fibrosis and left ventricular dysfunction after TAC. We further demonstrated that the improved cardiac function by Musclin is attributed in part to an increased cardiomyocyte contractility: While Musclin alone had no effect, it potentiated the CNP mediated inotropic response in isolated adult WT murine ventricular cardiomyocytes, which was, however, completely abrogated in cardiomyocytes lacking NPR-B. Although we cannot exclude additional effects of Musclin via enhancing ANP

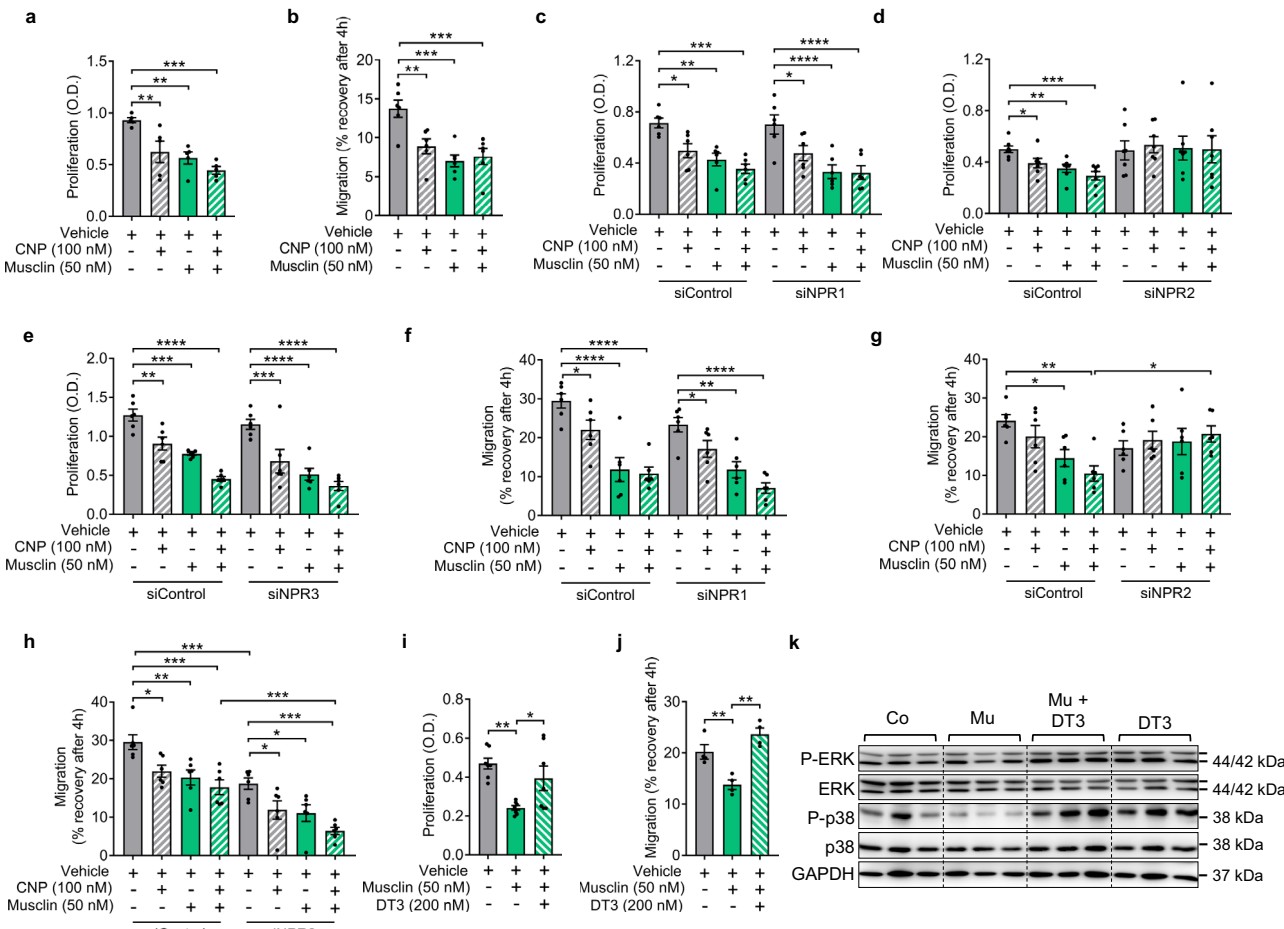

**Fig. 8 Musclin inhibits the activation of cardiac fibroblasts through NPR-B dependent activation of protein kinase G (PKG). a** Measurement of cardiac fibroblast proliferation by BrDU incorporation ELISA with addition of CNP and/or Musclin ($n = 5$/condition, **$p = 0.0039$ for vehicle vs. CNP, **$p = 0.0019$ for vehicle vs. musclin and ***$p = 0.0002$). **b** Assessment of cardiac fibroblast migration through detection of recovery of a scratch wound after 4 h during stimulation as indicated ($n = 6$/condition, **$p = 0.0023$, ***$p = 0.0003$ for vehicle-treated cells vs. stimulated with CNP, and ***$p = 0.0005$ vs. cells stimulated with CNP and Musclin). **c–e** Measurement of cardiac fibroblast proliferation by BrDU incorporation ELISA under the indicated conditions (**c–e**) ($n = 6$/condition, except siNPR2 treatment $n = 7$); *$p = 0.0103$ for siControl cells, vehicle treated vs. stimulated with CNP, **$p = 0.0016$ vs. stimulated with Musclin and ***$p = 0.0001$ vs. stimulated with CNP and Musclin, *$p = 0.0103$ for siNPR1 cells, vehicle treated vs. stimulated with CNP, ****$p < 0.0001$ vs. stimulated with Musclin and vs. stimulated with CNP and Musclin (**c**); *$p = 0.0243$, **$p = 0.0055$ and ***$p = 0.0003$ (**d**); **$p = 0.0021$ for siControl cells, vehicle treated vs. stimulated with CNP, ***$p = 0.0002$ vs. stimulated with Musclin, and ****$p < 0.0001$ vs. treated with both CNP and Musclin, ***$p = 0.0002$ for siNPR3 cells, vehicle treated vs. stimulated with CNP, ****$p < 0.0001$ vs. treated with Musclin and vs. cells treated with both CNP and Musclin (**e**). Assessment of cardiac fibroblast migration by scratch assay in cardiac fibroblasts ($n = 6$/condition) treated with siRNA, CNP and Musclin as indicated (**f–h**), *$p = 0.036$ for siControl cells, vehicle treated vs. treated with CNP, ***$p < 0.0001$ vs. Musclin and vs. stimulated with CNP and Musclin, *$p = 0.046$ for siNPR1 cells, vehicle treated vs. stimulated with CNP, **$p = 0.0013$ vs. Musclin and ****$p < 0.0001$ vs. cells treated with both CNP and Musclin (**f**); *$p = 0.0148$ for siControl cells, vehicle treated vs. treated with CNP, **$p = 0.0013$, and *$p = 0.0148$ between siControl and siNPR2 cells treated with both CNP and Musclin (**g**); *$p = 0.0162$ for siControl cells, vehicle treated vs. cells stimulated with CNP, **$p = 0.0042$ vs. cells treated with Musclin and ***$p = 0.0004$ vs. cells stimulated with both CNP and Musclin, *$p = 0.0162$ for siNPR3 cells, vehicle treated vs. stimulated with CNP or Musclin and ***$p = 0.0002$ vs. stimulated with Musclin and CNP, ***$p = 0.0005$ for siControl cells treated with Musclin und CNP vs. siNPR3 cells stimulated with Musclin and CNP, ***$p = 0.0008$ for vehicle-treated siControl cells vs. vehicle-treated siNPR3 cells (**h**). **i**, **j** Assessment of cardiac fibroblast proliferation and migration after stimulation with Musclin, CNP or the PKG inhibitor DT3 as indicated (for proliferation $n = 7$/condition, **$p = 0.0018$ for vehicle vs. Musclin treated cells and *$p = 0.016$ for cells stimulated with Musclin vs. cells treated with Musclin and DT3 (**i**), and for migration $n = 4$/condition, **$p = 0.0043$ for vehicle-treated cells vs. stimulated with Musclin, ***$p = 0.0005$ for cells treated with Musclin vs. treated with DT3 (**j**)). **k** Immunoblot from cardiac fibroblasts' protein lysate for the indicated proteins after stimulation as indicated. GAPDH was used as loading control. The size of the proteins is indicated in kDa. Data in bar graphs are shown as mean ± standard error of the mean (SEM). *$p < 0.05$, **$p < 0.01$, ***$p < 0.001$, ****$p < 0.0001$ determined by one-way ANOVA followed by the Holms–Sidak's multiple comparisons test. Mu stands for Musclin. Source data are provided as a source data file.

and BNP abundance in vivo, we showed that cardiomyocytes exhibited no inotropic response to Musclin following ANP stimulation in vitro, suggesting that Musclin augments cardiomyocyte contractility via enhanced CNP/NPR-B, but not ANP/NPR-A signaling. Interestingly, the augmenting effects of CNP on

cardiomyocyte contractility by Musclin were mimicked by treatment with cilostamide (a PDE3 inhibitor), which prevented further contractility enhancement by Musclin, indicating that PDE3 serves as a critical downstream target. By contrast, inhibition of the cGMP dependent protein kinase (PKG) with a selective inhibitor (DT3[46]),

did not prevent additional inotropic responses by Musclin during CNP stimulation, suggesting a predominant role of PDE3 compared to PKG under these circumstances. Importantly, we found that co-stimulation with Musclin and CNP increased cardiac cGMP and cAMP responses depending on the presence of NPR-B, whereas the lack of Musclin in KO mice reduced these responses. PDE3 binds both cAMP and cGMP, but cGMP acts as competitive inhibitor of cAMP hydrolysis, thereby creating a positive cGMP-to-cAMP crosstalk[47]. Pre-treatment with the PDE3 inhibitor cilostamide prevented a further cAMP increase by Musclin following CNP stimulation. Together, our data suggest that Musclin augmented cardiomyocyte contractility and relaxation by promoting CNP signaling through NPR-B/cGMP-mediated PDE3 inhibition, a positive cGMP-to-cAMP crosstalk and enhanced cAMP/PKA signaling. In support of our conclusions, a recent study demonstrated that CNP, but not BNP, enhanced the positive inotropic and lusitropic effects of adrenergic stimulation in rat cardiac muscle strips through the inhibition of PDE3[42]. As similar levels of cGMP were generated by BNP and CNP, the differences in downstream signaling were attributed to separate microdomains associated either with NPR-A or NPR-B dependent signaling[48,49]. In addition, NPR-A-dependent signaling becomes downregulated, while NPR-B accounts for the majority of the natriuretic peptide response in TAC-induced heart failure in mice[50]. Interestingly, the enhancement of cGMP-to-cAMP crosstalk was recently shown to account for compensation of cardiac function during pressure overload, before overt heart failure develops in mice[51]. We show here that skeletal muscle-derived Musclin employs a similar path to promote cardiac compensation. Although pharmacological PDE3 inhibition enhances contractility in rodents as well as in patients[52,53], treatment over longer periods increased the incidence of arrhythmias and sudden cardiac death[54]. While this might create a caveat against treatment approaches with Musclin, on the other hand new pharmacological interventions are sought that target PDE3 without increasing the risk of cardiac death[55]. Musclin uses a more indirect mechanism of PDE3 inhibition by enhancing protective CNP signaling, which additionally leads to potent anti-fibrotic, but - in contrast to direct PDE3 inhibition - does not trigger prohypertrophic effects or sudden death in mice. Therefore, Musclin might constitute such a new pharmacological intervention, although more studies are needed to validate this in large animal models of heart failure[55,56]. Miyazaki et al. reported that high dose intravenous injection or overexpression of Musclin in the liver of transgenic mice improved survival and attenuated cardiac hypertrophy, but did not improve heart function following myocardial infarction, mainly by reducing myocardial inflammation and infarct size[17]. While we did not investigate cardiac inflammation in our model, because it might play a less prominent role during pressure overload, we did not observe major effects of Musclin (neither through overexpression, nor through its ablation) on cardiac hypertrophy. This discrepancy could either be due to the different heart failure models, and/or due to the different extent of overexpression used in both studies: with regard to plasma levels, Miyazaki et al. employed a more than ten-fold overexpression, while in our case overexpression was about two-fold. In addition, Musclin might have been abnormally processed due to its unphysiological expression in the liver in the study by Miyazaki et al.[57]. Beside its positive inotropic effects, skeletal muscle Musclin potently inhibited myocardial fibrosis during TAC, which likely also contributed to improved heart function. Mechanistically, our data suggest that Musclin (as well as CNP) act on cardiac fibroblasts to oppose their activation by inhibiting cellular proliferation and migration. In contrast to what we observed in cardiomyocytes, there were similar effects on fibroblasts by CNP and Musclin, but almost no additive effect. We hypothesize that this is because unlike cardiomyocytes, fibroblasts produce CNP by themselves and it is therefore already present when Musclin is added to act in synergy, although this could not be enhanced by additional external CNP. Musclin dependent inhibition of fibroblast activation was blunted by siRNA mediated NPR-B downregulation, indicating that it also acts in fibroblasts via enhancing CNP/NPR-B signaling. Cardiac fibroblast activation and fibrosis are mediated by the MAP kinase p38[38]. We found here that Musclin inhibits p38 activation and that PKG was necessary for Musclin dependent fibroblast and p38 inhibition. In this regard, anti-fibrotic and p38 inhibiting effects by PKG were previously described, suggesting a Musclin/CNP/NPR-B/PKG/p38 dependent pathway in cardiac fibroblasts[35–37,58].

Musclin is a known high affinity ligand of NPR-C, which has been traditionally viewed as clearance receptor for the natriuretic peptides. Indeed, we found that Musclin and CNP compete for binding to NPR-C and that Musclin administration elevated CNP levels in the supernatant of NPR-C expressing Hek293 cells. In addition, more or less plasma CNP concentrations were found in skeletal muscle-specific Musclin overexpressing or KO mice, respectively, which is compatible with previous studies demonstrating increased systemic CNP levels in response to enhanced Musclin abundance[16,17]. Importantly, there is strong evidence emerging from *Npr3* KO mice that direct NPR-C dependent signaling (for example involving Gi proteins and ERK signaling) could be responsible for the protective roles of CNP[59–61]. In our study, however, Musclin/CNP co-stimulation triggered increased cAMP levels in cardiomyocytes (as opposed to decreased cAMP levels that would have been expected due to NPR-C-Gi signaling) and also did not affect ERK signaling in cardiac fibroblasts. Moreover, a vital cardioprotective role was recently demonstrated for the CNP receptor NPR-B, as heterozygous *Npr2* KO mice develop heart failure with myocardial fibrosis in advanced age[62]. Similarly, cardiomyocyte-specific ablation of NPR-B, triggered cardiac dysfunction after TAC[63]. This, together with the abrogation of Musclin effects in cardiomyocytes and fibroblasts by NPR-B ablation/downregulation, argues against a major role of direct NPR-C signaling in mediating the effects of Musclin, although our study was not optimally designed to clarify this point. The role of NPR-C for the cardiac Musclin effects in vivo should be evaluated in *Npr3* KO mice in the future. In addition, it is possible that Musclin acts (at least in part) via a currently unknown (co-) receptor.

Therapeutic overexpression of Musclin in skeletal muscle might also be beneficial in patients: Musclin (*OSTN*) mRNA levels in skeletal muscle biopsies from patients suffering from heart failure and sarcopenia or cachexia were markedly downregulated. In addition, patients with heart failure due to aortic stenosis exerted reduced Musclin serum levels, although it was not clear whether these patients suffered from sarcopenia or cachexia. Future studies of larger patient cohorts will be necessary to address whether Musclin is suitable as biomarker. Intriguingly, physical activity in patients suffering from heart failure is emerging as therapeutic approach, which might be, at least in part, the result of an enhanced secretion of protective myokines such as Musclin, which is upregulated in skeletal muscles by exercise[64,65,13]. In conclusion, our observations provide insights into the role of myokines during skeletal muscle wasting and heart failure. Musclin, a skeletal muscle derived and secreted myokine, attenuates cardiac dysfunction and myocardial fibrosis by augmenting the CNP/NPR-B stimulated crosstalk of cGMP to cAMP in cardiomyocytes and by inhibiting p38 MAPK signaling through the activation of PKG in cardiac fibroblasts. Enhancing inter-tissue communication via AAV6-mediated overexpression of the myokine Musclin could be a promising therapeutic strategy for treating heart failure progression.

## Methods

**Animal experiments**. All procedures involving the use and care of animals were performed according to the Directive 2010/63/EU of the European Parliament on the protection of animals used for scientific purposes and the German animal protection code. Approval was granted by the local state authorities (LAVES-Niedersächsisches Landesamt für Verbraucherschutz und Lebensmittelsicherheit, 33.14-42502-04-11/0335 and 33.8-42502-04-16/2356).

Male C57Bl/6JCrl mice (Charles River Laboratories) at 8–10 weeks of age were used for all experiments (except for the experiments involving Musclin KO and littermate WT mice, as well as Npr1 and Npr2 KO and corresponding WT mice, where male and female mice were used). The animals had free access to water and a standard diet and were maintained on a 12-h light and dark cycle at a room temperature of 22 ± 2 °C and a humidity of 35–60%. TAC surgery was performed by subjecting the aorta to a defined 27-gauge constriction in 8–10 weeks old mice as described previously[66,67]. Cardiomyocyte specific, 8–10 weeks old Npr1 (NPR-A) KO and global Npr2 (NPR-B) KO mice were previously published[68,69].

Echocardiography was performed with a linear 30 MHz transducer (Vevo 770 and Vevo3100, Visualsonics, Toronto, Canada) in mice that were sedated with 2% isoflurane. LV enddiastolic area, enddiastolic average wall thickness (Wth) and enddiastolic volume (EDV) as well as endsystolic area (LVESA) and endsystolic volume (ESV) were recorded or calculated. Ejection fraction was calculated as [(EDV–ESV)/EDV] × 100. The degree of pressure overload was assessed utilizing the ratio of right over left common carotid artery blood flow peak velocity (RCA and LCA, resp.), which were each recorded with pulsed-wave Doppler at the level of the carotid bifurcation. Left ventricular pressure was assessed with a microtip pressure-volume catheter (PVR-1000, Millar Instruments, Houston, TX) inserted through the cardiac apex during anesthesia with 2% isoflurane. Tail cuff blood pressure was measured in awake mice (BP 2000 Blood Pressure Analysis System, Visitech). Premature death was a criterion for exclusion from an ongoing experiment.

MRI in mice was performed using a 7-T small-animal MRI unit (Bruker, Pharmascan, PHS70/16US, Ettlingen Germany) and a circular polarized volume coil (Bruker T10327V3). Animals were anesthetized by using isoflurane inhalation (1–2% in oxygen). A respiratory-triggered T1-weighted RARE sequence covering the entire abdomen in transverse orientation was used to evaluate intra-abdominal and subcutaneous fat volumes as well as autochthone muscle area. Sequence parameters were: repetition time 1000 ms, echo time 20.64 ms, field of view 30 mm × 35 mm, matrix size 176 × 194 (resolution 0.17 × 0.18 mm), slice thickness 1 mm. In order to identify fat tissue, images were acquired with and without fat saturation. Intra-abdominal and subcutaneous fat volumes were quantified by semi-automatic segmentation on a length of 2 cm from head to feet, beginning from the upper pole of the right kidney using a region growing algorithm and Horos v1.1.7 software (https://horosproject.org/). In addition, autochthone muscle area was quantified at the level of the right kidney hilum using a region of interest.

**AAV6-mediated overexpression of Musclin in vivo**. The entire coding region of the Musclin cDNA was Myc-tagged and subcloned into the pdsMCKE vector downstream of the muscle creatine kinase promoter. The resulting plasmid was used to produce AAV6 Musclin vectors. AAV6 vectors expressing renilla luciferase downstream of the same promotor were generated and used as control (AAV6 Control). AAV6 vectors ($5 \times 10^{11}$ viral genomes in NaCl, 0.9%) were injected intramuscular into the right quadriceps muscle of 8 weeks old male C57Bl/6JCrl mice 2 days after TAC or sham surgery.

**Generation of skeletal muscle-specific and inducible Musclin deficient mice**. Vector construction and targeted KO strategy were designed together with the InGenious Targeting Laboratory. Briefly, a conditional loxP KO mouse (Ostn$^{fl/fl}$) with homologous recombination targeting the exon 3 (209 bp) was generated, which results in a frame shift upon deletion.

The ~10.4 kb region used to construct the targeting vector was first subcloned from a positively identified C57BL/6 fosmid clone (WI1-575B23) using a homologous recombination-based technique. The region was designed such that the 5′ homology arm extends 7.0 kb 5′ to the single LoxP site. The 3′ homology arm extends 2.7 kb downstream of the LoxP/FRT flanked Neo cassette. The 5′ LoxP site was inserted about 83 bp upstream of exon 3 and the 3′ LoxP site along with the FRT flanked Neo selection cassette was placed downstream of exon 3. The Neo cassette is about 303 bp away from exon 3. The size of the target region is 595 bp containing exon 3. The targeting vector was confirmed by restriction analysis after each modification step, and by sequencing using primers designed to read from the selection cassette into the start of the 3′ arm (IVNeoSQ3) and the 3′ end of the targeting region (IVNeoSQ7). The targeting construct was linearized using NotI prior to electroporation into ES cells. Targeted iTL BF1 (C57BL/6 FLP) embryonic stem cells were microinjected into Balb/c blastocysts. Resulting chimeras with a high percentage black coat color were mated to C57BL/6 WT mice to generate Germline Neo Deleted mice. PCR and Southern blot analysis verified Musclin targeting in the offspring of F1 and F2 mice.

Next, to generate inducible Musclin KO mice, Ostn$^{fl/fl}$ mice were crossed with a HSA-rtTA/TRE-Cre double transgenic mice (The Jackson Laboratory) that express Cre recombinase only in skeletal muscle and only following Dox treatment (Ostn$^{fl/fl}$ (HSA-rtTA/TRE)Cre). The first transgene uses the HSA promoter to drive expression

of the rtTA. The second transgene uses a tetracycline responsive promoter to drive the expression of Cre recombinase[70].

For experiments, Dox (Sigma Aldrich) was dissolved in drinking water at a concentration of 2 mg/ml and given to animals in light-protected water bottles for one week before TAC or sham surgery. Littermate male and female Ostn$^{fl/fl}$ mice (Musclin WT) on a C57Bl/6 J background as control and Ostn$^{fl/fl}$ (HSA-rtTA/TRE)Cre mice (Musclin KO), aged between 8–10 weeks, all treated with Dox were included in this study.

**RNA isolation and quantitative real-time PCR**. Total RNA from hearts or muscles was isolated with the Trifast reagent (Peqlab, Erlangen, Germany). cDNA was generated from 1 µg RNA using Maxima H Minus First Strand cDNA Synthesis Kit (Thermo Fisher Scientific) and quantitative PCR (from of 10 ng mRNA equivalent) was performed using standard procedures with the MX4000 multiplex qPCR system from Stratagene. Gene expression was normalized to Gapdh mRNA expression. Absolute quantification of Ostn (Musclin) mRNA was based on a standard curve, which was prepared with a plasmid containing the Musclin sequence. Primer sequences can be found in Supplementary Table 3.

**Deep sequencing and bioinformatics**. To identify genes regulated in cachectic skeletal muscle, RNA was isolated from the quadriceps muscle 12 weeks after sham or TAC surgery using the RNeasy mini kit (Qiagen) following the instruction manual. Further analysis was performed at the Helmholtz Center for infection research in Braunschweig. Quality and integrity of total RNA was controlled on the Agilent Technologies 2100 Bioanalyzer (Agilent Technologies; Waldbronn, Germany). The RNA-sequencing library was generated from 100 ng total RNA using Dynabeads® mRNA DIRECT™ Micro Purification Kit (Thermo Fisher) for mRNA purification followed by ScriptSeq v2 RNA-Seq Library Preparation Kit (Epicenter) according to the manufacturer's protocols. The libraries were sequenced on Illumina HiSeq2500 using TruSeq SBS Kit v3-HS (50 cycles, single ended run) with an average of $3 \times 10^{7}$ reads per RNA sample. Before alignment to the reference (mm10) each sequence was trimmed on base call quality and sequencing adapter contamination using the Trim Galore! wrapper tool (Babraham Bioinformatics—Trim Galore!). Reads shorter than 20nt were removed from FASTQ files. Trimmed reads were aligned to the reference using the short read aligner STAR (https://code.google.com/p/rna-star/). Feature counts were determined using R package "Rsubread". Only genes showing counts greater than 5 at least two times across all samples were considered for further analysis (data cleansing).

Differentially regulated genes between sham and TAC ($1 \leq \log2FC \leq -1$) with p < 0.05 were identified by EdgeR. GO (Biological Process) analysis was performed separately for up- and downregulated genes using the DAVID tool (DAVID 6.8: https://david.ncifcrf.gov/). Identified transcripts were filtered for genes assigned to the GO classification "Cellular Compartment" "Extracellular Region" (GO:0005615) from (http://www.informatics.jax.org/vocab/gene_ontology). The resulting transcripts were again filtered for genes with higher expression in the quadriceps muscle (quadriceps muscle vs. in adult heart (log2FC > 2) with p < 0.05) and identified by EdgeR.

**Protein extraction and immunoblotting**. Protein was extracted from frozen heart or muscle tissue using lysis buffer (10 mM Tris, 150 mM NaCl, 4% Glycerol, 0.5 mM Sodium metabisulfite, 1% Triton X, 0.1% Sodium deoxycholate, 0.05% SDS, pH 7.5) with freshly added Protease Inhibitor Cocktail (#11697498001, Roche) and Phosphatase Inhibitor Cocktail (#524629, Merck). Protein concentration was determined utilizing a Pierce BCA assay kit (#23225, Thermos Scientific). Equal amounts of total protein (20 µg) were separated on SDS-polyacrylamide gel under reducing conditions and transferred to PVDF membranes (#IPVH00010, Merck). Immunoblot analysis was conducted using standard techniques with antibodies to the following proteins: Musclin (#RD181079100, 1 µg/ml, Bio-Vendor), Actin (#A2066, 1:100, Sigma), alpha Tubulin (#ab40742, 1:5000, Abcam) and Serca2a ATPase (#ab2861, 1:1000, Abcam), Phospholamban (#A010-14, 1:2000, Badrilla) and Phospholamban (pSer16) (#A010-12, 1:5000, Badrilla), p38 MAPK (#9212, 1:1000, Cell Signaling), Phospho-p38 MAPK (T180/Y182), (#9211, 1:1000, Cell Signaling), p44/42 MAPK (Erk1/2), (#9102, 1:1000, Cell Signaling), Phospho-p44/42 MAPK (T202/Y204), (#9101, 1:1000, Cell Signaling). The blots were hybridized with an antibody against GAPDH (#10R-G109a, 1:6000, Fitzgerald) to verify equal loading of protein in each lane. The following secondary antibodies were used: Anti-Rabbit-IgG-HRP (#NA934, 1:3500, GE Healthcare Life Sciences), anti-Mouse-IgG-HRP (#NXA931, 1:3500, GE Healthcare Life Sciences), and anti-Goat-IgG-HRP (#HAF005, 1:10000, R&D systems).

**Myocyte Isolation, cell contractility, sarcomere length, and $Ca^{2+}$ transient measurements**. Adult ventricular cardiomyocytes were prepared from mice as previously described using a Langendorff system[71]. Following isolation, the ventricular myocytes were placed on laminin-coated glass coverslips (24 × 24 mm, Roth), then 3 h later washed twice with long-term incubation medium (200 mg BSA, 1 ml 1 M HEPES, 1x Penicillin, 1× L-Glutamin, 100 µl ITS 1000× in 97.9 ml MEM medium). The isolated myocytes were transferred to the recording chamber of the IonOptix System. The cells were stimulated with recombinant C-type natriuretic peptide (CNP, #N8768-1MG, Sigma Aldrich), Musclin (#ab105614), cilostamide (#BML-PD125-0005, Enzo) or the

cell permeable PKG Iα inhibitor DT3 (#370655, Merck). Sarcomere shortening was assessed upon field stimulation (1 Hz) using a video-based sarcomere length detection system (IonOptix Corporation) at 37 °C. The recordings were subsequently analyzed with the Ion Wizard software (IonOptix). For measurement of $Ca^{2+}$ transients in intact cardiomyocytes, cells were incubated in modified Tyrode's solution (in mmol/L: 135 NaCl, 4.7 KCl, 0.6 $KH_2PO_4$, 0.6 $Na_2HPO_4$, 1.2 $MgSO_4$, 1.25 $CaCl_2$, 20 glucose, 10 Hepes, pH 7.46) containing 1 µmol/L Fura-2-AM for 20 min. Cells were alternatively excited at 340 and 380 nm with 510 nm emissions using the hyper-switch dual excitation light source (IonOptix). The F340/F380 ratio was used as an index of cytosolic $Ca^{2+}$ concentration.

**cAMP and cGMP measurements by FRET**. For FRET-based measurements of cAMP and cGMP, cardiomyocytes from either cGES-DE5 transgenic (for cGMP measurements) or from Epac2-camps (for cAMP measurements) transgenic mice were isolated[32,33]. Thereafter, the cells were washed once and maintained in a physiological buffer containing: NaCl 144 mM, KCl 5.4 mM, $CaCl_2$ 1.0 mM, $MgCl_2$ 1.0 mM, and HEPES 10 mM, pH 7.4 at room temperature. Cells were placed on a standard inverted microscope (Leica DMI3000B) equipped with 60x/1.5oil immersion objective, single-wavelength light-emitting diode (CoolLED pE-100, 400 nM or 440 nM), DV2 DualView beam splitter (Photometrics) and OptiMOS charge couple device (CCD) camera (QImaging). Cells were consecutively stimulated with the indicated substances and the GFP/RFP (for cGMP measurements) or CFP/YFP (for cAMP measurements) emission ratio upon 400 nm (for cGMP measurements, filters GFP 520 ± 30, RFP 630 ± 50) or 440 nm excitation (for cAMP measurements, filters YFP 535 ± 40 nm, CFP 480 ± 30 nm) was measured. After each measurement, emission values were corrected for bleed through of GFP into RFP or of CFP into YFP channel and for photobleaching as described[72].

**Fibroblast isolation and siRNA treatment**. Adult murine cardiac fibroblasts were isolated from 3–5 weeks old CD-1 mice. The hearts were cut into small pieces and digested with Liberase TH (#5401151001, Roche) in SADO-Mix Solution (20 mM HEPES-NaOH (pH 7.6), 130 mM NaCl, 3 mM KCl, 1 mM $NaH_2PO_4$, 4 mM Glucose, 1.5 mM $MgSO_4$ in $ddH_2O$). The harvested cardiac fibroblasts were incubated in DMEM cell culture media containing 10 % FCS upon reaching confluence. siRNA treatment using control siRNA (Ambion) or siRNA directed against Npr1, Npr2 or Npr3 (Dharmacon) was conducted at a final concentration of 100 nM using Lipofectamine (Invitrogen). The siRNA sequences used are listed in Supplementary Table 4. Fibroblast proliferation and migration assays were conducted 48 h after transfection.

**Fibroblast migration assay**. A scratch assay was performed to study fibroblast migration. The cells were plated on 24-well plates. Media was added containing 10% FCS with or without additional agents. Twenty-four hours later, the cell monolayer of growing fibroblasts was scraped in a straight line to create a "scratch-injury" with a p200 pipet tip. Pictures were taken immediately after scratch and 4 h later. The scratch-injury area was analyzed using AxioVision 4.8 Software.

**Fibroblast proliferation assay**. DNA synthesis was measured by performing a BrdU incorporation assay with a commercially available kit (Roche). Fibroblasts were plated on 96-well plates in 10 % FCS culture media with or without the addition of different agents. Four hours later, BrdU was added. BrdU incorporation was determined by Cell Proliferation ELISA (#11647229001, Roche) 24 h later following the manufacturer's protocol.

**NPR-C competition assay**. Wells of a 96-well high-binding plate (Costar #2592) were coated with 100 ng of recombinant mouse NPR-C/NPR3 Fc Chimera protein (R&D Systems, #10187-NR). After washing and blocking of the wells with 1%BSA/PBS, 100 µl of 200 ng/ml Musclin (R&D System, #9700-ON) alone or with increasing amounts of CNP (Sigma, #N8768) were added to the coated wells and incubated overnight at 4 °C. On the next day, after several washes, the wells were incubated with anti-Musclin antibody (R&D Systems, #10187-NR) and an HRP coupled secondary antibody (R&D Systems, #HAF0169) to detect Musclin that was bound to the NPR-C coated well. Relative Musclin binding was calculated with the help of a Musclin standard curve.

HEK293 cells (Leibniz Institute DSMZ, Cell Line 293 ACC 305) were transfected with NPRC-GFP (OriGene, #MG208587, Accession No. NM_008728) or were left untransfected. The cells were stimulated with recombinant C-type natriuretic peptide (CNP, #N8768, Sigma Aldrich) with or without the addition of Musclin (#RD172079100, BioVendor) for 1 h. The CNP levels were determined in the supernatants using the CNP ELISA kit (#MBS2514795, detecting mature CNP peptides, MyBioSource) according to the manufacturer's protocol.

**Measurement of cGMP, cAMP, ANP, CNP, and Musclin by ELISA**. cGMP levels were measured using the cGMP enzyme immunoassay (EIA) kit (#581021, Cayman Chemical). Isolated adult murine cardiomyocytes were homogenized in 100 µl of 0.1 M HCl and centrifuged at $1000 \times g$ for 10 min. The supernatants were diluted 1:6 with diluent buffer and used for cGMP level measurements according to the manufacturer's protocol. For the measurement of cAMP levels using the cAMP enzyme

immunoassay (EIA) kit (#581001, Cayman Chemical) the samples were processed as described above. The plasma Musclin concentration was determined by the mouse Osteocrin ELISA (#EKC37526, Biomatik), the ANP concentrations in plasma by the ANP-Fluorescent EIA kit (Phoenix Pharmaceuticals, #FEK-005-24) and the CNP plasma concentration using the CNP ELISA kit (#MBS2514795, detecting mature CNP peptides, MyBioSource) according to manufacturer's instructions.

**Analysis of human material**. Muscle biopsies of the vastus lateralis muscle were obtained from control subjects without history of cardiovascular or neuromuscular disorders. They were enrolled at the Department of Cardiology, Charité Medical School, Campus Virchow-Klinikum, Berlin, Germany, and the Space Clinic of the Institute of Space Medicine and Physiology (Medes-IMPS, Rangueil Hospital) in Toulouse, France. Individuals enrolled at the Institute of Space Medicine and Physiology received a compensation for their participation. All subjects provided written and informed consent before being enrolled. Sarcopenic and cachectic patients were enrolled in the Studies Investigating Co-morbidities Aggravating Heart Failure (SICA-HF) trial and were enrolled at the Department of Cardiology, Charité Medical School, Campus Virchow-Klinikum, Berlin, Germany[7,73]. Sarcopenia was defined as an Appendicular Muscle Mass (kg)/height $(m)^2 < 5.45$ in women and $<7.26$ in men. The local ethics committees at the Charite Medical School, Germany, and at the CPP Sud-Ouest et Outre-Mer I, France, approved the studies. Cachexia was defined as non-edematous, non-intentional weight loss ≥5% over a period of 1 year. Patient and control characteristics are shown in Table 1. The study fulfills all principles of the Declaration of Helsinki. Human serum Musclin concentrations were determined by the human Musclin ELISA kit (Cusabio, # CSB-E12021h) according to manufacturer's instructions in male healthy blood donors or male patients suffering from severe aortic stenosis before aortic valve replacement was conducted (the patient characteristics are shown in Table 2). This study was approved by the Ethical Committee of the Hannover Medical School, Germany. All individuals and patients gave written informed consent.

**Histology**. After removal from the chest cavity, the hearts were rinsed in PBS and 0.5 M KCL. Transversal frozen sections (7 µM in thickness) of the myocardium were generated. To measure cardiomyocyte dimensions, the cardiomyocyte cell membranes were stained with TRITC-WGA (20 µg/ml, Sigma Aldrich) and the nuclei with 4',6-diamidino-2-phenylindole (#HP20.1 ready-to-use DAPI, ROTH). Fluorescein-labeled GSL I-isolectin B4 (10 µg/ml, IB4, Vector Laboratories) was used to visualize cardiac endothelial cells. The cellular size of cardiomyocytes in situ was determined with Image J software.

Fibrosis was quantified with the Sirius Red staining method in 12-µM-thick cryosections. Subsequently, the fraction of the fibrotic area (stained in red) from the total myocardial area was determined using Adobe Photoshop Imaging Software. Representative images of histological sections and immunofluorescence stainings are shown. Muscles were dissected, embedded in a drop of OCT compound and frozen in 2-methylbutane precooled in liquid nitrogen. Cryosection (7 µm thickness) of the quadriceps muscle was generated and stained with tetramethyl rhodamine isothiocyanate-conjugated wheat-germ agglutinin (TRITC-WGA, Sigma Aldrich) for the quantification of muscle fiber size.

For ATPase staining, 7 µm cryosections were exposed to pre-incubation solution (50 mM sodium acetate, 30 mM sodium barbital, brought to pH 4.2 with HCl) at pH 4.2 for 5 min. This was followed by a 30 min incubation in 20 mM sodium barbital, pH 9.4, containing 9 mM $CaCl_2$ and 2.7 mM ATP at 37 °C, before incubations with 2% $CoCl_2$ and then $1\%(NH_4)_2S$ were conducted and the slices were embedded after several washes as described[74].

For Musclin immunostaining of quadriceps muscles, 7-µm-thick cryosections were fixed with 100 % ice-cold acetone for 20 min at room temperature, dried, and permeabilized with PBS containing 0.3% Triton X-100 for 20 min, before they were blocked with 3% BSA in PBS for 30 min. The cryosections were then incubated overnight at 4 °C with the primary antibody against Musclin (1:100, # RD181079100, Biovendor), followed by the AlexaFluor 488-coupled secondary antibody (#4412 S, 1:500, Cell Signaling) for 4 h in the dark. Nuclei were stained with 4',6-diamidino-2-phenylindole (#HP20.1 ready-to-use DAPI, ROTH). Coverslips were mounted on glass slides with VECTASHIELD HardSet Mounting Medium (Vector Laboratories).

**Statistics**. For statistical analysis GraphPad Prism software (version 7.04, GraphPad Software Inc.) was used. Data are shown as mean ± standard error of the mean (SEM) unless noted otherwise in the legend. All experiments were carried out in at least three biological replicates. The exact number of biological replicates (number of mice, samples, or cell culture dishes) is indicated in the figure legends. Sample size was chosen as a result of previous experience regarding data variability in similar models and experimental set-ups[66,67,75,76]. No statistical method was used to predetermine sample size. Mice were allocated to the different experimental group due to genotyping results. WT mice were randomly assigned to receive AAV6 Control or AAV6 Musclin treatment. Healthy individuals, patients suffering from severe aortic stenosis or from CHF with cachexia or sarcopenia were assigned to the experimental groups primarily as result of their diagnosis (or due to absence of disease for healthy individuals). For the serum analysis all samples (randomly collected during between May 2011 and February 2014 among patient with severe aortic stenosis or from blood donors) available in our lab were analyzed and

included in the manuscript. For the skeletal muscle study, samples in each group were randomly chosen from the existing samples of the group. The investigators were blinded for mouse genotype and treatment during surgeries, echocardiography, cardiac catheterization, organ weight determination and all histological and immunofluorescence quantifications. For other experiments, researchers were not blinded to group allocation during data collection due to the necessity of knowing the treatment to be administered or the samples to be collected. However, researchers were blinded to group allocation during data analysis in most of the cases. The variance was comparable between groups. The Shapiro–Wilk normality test was conducted to assess normality. If data were normally distributed, for unpaired data, multiple groups were compared by one-way measures analysis of variance (ANOVA) followed by the Holms–Sidak's multiple comparisons test or by unpaired, two-sided Student's $t$ test when comparing two experimental groups. For paired data, multiple groups were compared by one-way repeated measures ANOVA followed by the Holms–Sidak's multiple comparisons test or by paired, two-tailed $t$ test when comparing two experimental groups. When $n = 3$/condition, or when the normality test was negative, the Kruskal–Wallis test with Dunn's multiple comparisons test or the two-tailed Mann–Whitney test (in the case of only two experimental groups) were used as non-parametric test. A two-tailed $p$ value of <0.05 was considered significant.

**Reporting summary**. Further information on research design is available in the Nature Research Reporting Summary linked to this article.

## Data availability

All data presented in this study are available in the main text, the main figures or in the supplementary materials. RNAseq data generated during this study are available in the Gene Expression Omnibus (GEO) repository (http://www.ncbi.nlm.nih.gov/geo/) and are accessible through GEO series accession number GSE129205. The following genome assembly was used for alignment: mm10, GRCm38 - mm10 - Genome - Assembly - NCBI (nih.gov). Source data are provided with this paper.

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

## Acknowledgements

This study was supported by the Deutsche Forschungsgemeinschaft through the Cluster of Excellence Rebirth (EXC 62/1 and EXC 62/3 to K.C.W., J.B. and J.H.), the Heisenberg Program (HE 3658/6-1&2 to J.H.) and Research Grants (HE 3658/16-1 to J.H., EL 270/7-1 to A.E. and WA 2586/4-1 to M.W. and KU 1037/6-1 to M.K.). J.H., T.T., K.C.W., and J.B. were supported by the KFO311 (DFG). J.H. was also supported by the SFB1366/1 (project A06). B.K. was supported by a research grant of the German Cardiac Society (DGK) and by the Hannover Medical School ("Hochschulinterne Leistungsförderung"-HiLF). We wish to thank the Institute of Pathology of the Hannover Medical School for their expert technical assistance.

## Author contributions

M.S. designed experiments, conducted the majority of the experiments, acquired and analyzed data, and helped to prepare the first draft of the manuscript. B.K. designed experiments, conducted experiments, acquired and analyzed data, and prepared the first draft of the manuscript. A.M.G., F.A.T., G.M.D., A.G., A.A., K.D., M.M., T.H., and M.K.K. acquired and analyzed data. K.V. conducted experiments. T.G.M., C.P.T., and N.H. acquired and analyzed data. N.F. designed experiments and critically edited the manuscript. C.Z. analyzed the patient data. M.M.M. conducted experiments and analyzed data. X.L., M.W., and A.E. provided reagents, critically supported the contractility measurements, and revised the manuscript. R.G., J.C., and G.D. acquired and analyzed RNAseq data and critically edited the manuscript. S.B. and T.T. acquired and analyzed data and critically edited the manuscript. N.B. and V.O.N. acquired and analyzed data and critically edited the manuscript. O.J.M. and H.A.K. provided crucial reagents and critically edited the manuscript. J.S., J.F., M.B., and S.v.H. provided crucial patient samples and data and critically edited the manuscript. T.K., K.C.W., M.K., and J.B. provided experimental support and advice for the study and critically edited the manuscript. J.H. initiated and designed the study, designed and conducted experiments, analyzed data, supervised all experiments, and prepared the final draft of the manuscript.

## Funding

## Competing interests

The authors declare no competing interests.

## Additional information

[1]Department of Cardiology and Angiology, Hannover Medical School, Hannover, Germany. [2]Department for Cardiovascular Physiology, European Center of Angioscience (ECAS), Medical Faculty Mannheim, Heidelberg University, Mannheim, Germany. [3]Institute for Diagnostic and Interventional Radiology, Hannover Medical School, Hannover, Germany. [4]Central Animal Facility, Hannover Medical School, Hannover, Germany. [5]Institute of Molecular and Cell Physiology, Hannover Medical School, Hannover, Germany. [6]Institute of Physiology and Comprehensive Heart Failure Center, University of Würzburg, Würzburg, Germany. [7]Department of Cardiology and Pneumology, University of Göttingen Medical Center, DZHK (German Center for Cardiovascular Research), partner site Göttingen, Göttingen, Germany. [8]Experimental and Clinical Research Center (ECRC), Charité-University Medical Center Berlin, Max Delbrück Center (MDC) for Molecular Medicine in the Helmholtz Association, Berlin, Germany. [9]Department of Cardiology, Heart Center Brandenburg and Medical University Brandenburg (MHB), Bernau, Germany. [10]Institute of Pharmacology and Toxicology, Dresden University of Technology, Dresden, Germany. [11]Department of Electrophysiology, Heart Center, Dresden University of Technology, Dresden, Germany. [12]Anatomy and Developmental Biology, ECAS, Medical Faculty Mannheim, Heidelberg University, Mannheim, Germany. [13]RG Genome Analytics, Helmholtz Center for Infection Research, Braunschweig, Germany. [14]Institute of Molecular and Translational Therapeutic Strategies, Hannover Medical School, Hannover, Germany. [15]National Heart and Lung Institute, Imperial College London, London, UK. [16]Excellence Cluster REBIRTH, Hannover Medical School, Hannover, Germany. [17]Institute of Experimental Cardiovascular Research, University Medical Center Hamburg-Eppendorf, Hamburg, Germany. [18]Department of Internal Medicine III, University Hospital Schleswig-Holstein, Kiel, Germany. [19]DZHK, partner site Hamburg/Kiel/Lübeck, Hamburg, Germany. [20]Department of Cardiology, Angiology, and Pneumology, Internal Medicine III, University Hospital Heidelberg, Heidelberg, Germany. [21]DZHK, partner site Heidelberg/Mannheim, Heidelberg, Germany. [22]Berlin Institute of Health at Charité – Universitätsmedizin Berlin, BIH Center for Regenerative Therapies (BCRT), Berlin, Germany. [23]Department of Internal Medicine B, Cardiology, University Medicine Greifswald, Greifswald, Germany. [24]DZHK, partner site Greifswald, Greifswald, Germany. [25]Present address: Department of Medicine, Cardiology, Goethe University Hospital, Theodor-Stern-Kai 7, 60590 Frankfurt, Germany. [26]These authors contributed equally: Malgorzata Szaroszyk, Badder Kattih. ✉email: Badder.Kattih@kgu.de; Joerg.Heineke@medma.uni-heidelberg.de

