## [Peer Review File · Nature Communications]

Reviewers' comments:

Reviewer #1 (Remarks to the Author):

The article titled, "Skeletal muscle derived Musclin protects the heart during pathological overload," by Kattih et al, describes murine studies investigating the protective role of Musclin in cardiac dysfunction. They found that Musclin knockout mice had exaggerated cardiac dysfunction while mice overexpressing of Musclin in muscle had attenuated cardiac dysfunction. More specifically, they performed TAC on mice to pressure overload the heart for 12 weeks and saw reduce leg muscle weights consistent with cachexia. They isolated RNA from the leg muscle of heart failed or control mice and performed RNAseq analysis. They found that about 800 genes were upregulated and about 800 genes were downregulated. One of the mRNAs that was downregulated was Musclin, which was downregulated 3.5-fold. They also determined that Musclin levels were decreased by 30% in human heart failure patients compared to healthy individuals, although this small decrease seems unlikely to have an impact on Npr1 and Npr2 signaling. When Musclin was knockout, heart failure consequences were exacerbated, but when Musclin was overexpressed by an AAV vector, heart failure was ameliorated. This is reasonable work, but some of the cardiac results have been previously reported (1). However, the most important new information is "how" the increased musclin levels improve heart function. Is it through Npr1, Npr2 or possibly Npr3? Unfortunately, the experiments that are designed to differentiate between these pathways rely on unproven inhibitors of Npr1 and Npr2. A much better approach is to selectively remove these receptors in specific tissues using Cre/Lox technology and ask if the effects go away, or at the very least, are markedly diminished, in mice where these receptors are not expressed in the tissues hypothesized to be critical for the physiologic effects of Musclin. Floxed mice for both Npr1 and Npr2 have been available for years.

Major issues:

A large amount of the data supporting this manuscript is based on so called selective inhibitors of Npr1, Npr2 and/or PKG. In this reviewer's opinion, there are no specific inhibitors of Npr1, Npr2 or Npr3. If the authors disagree, then please show data where Npr1 or Npr2 were expressed in NP receptor lacking cells that have been transfected with a single NP receptor, which is required to show which receptor is being activated. In this situation, is there data showing that an inhibitor inhibits one receptor and not the other? I am not aware of papers that report these critical data so please reference it. A much better and believable way to conduct a similar experiment is to use mice

with Lox P sites flanking the genes for Npr1 or Npr2 in order to specifically delete these receptors in specific cell types.

If the authors hypothesis is correct, the AAV directed expression of Musclin should dramatically increase ANP, BNP and CNP levels in these mice. Is this true?

Minor Comments:

It makes no sense to say that “Musclin is expressed almost exclusively in skeletal muscle,” and in the very next phrase say that it is also expressed in bone. In fact, the same protein was identified by a separate group and named osteocrin.

Musclin immunoblots are inconsistent and the differences look more like transfer issues rather than real differences in protein concentrations.

What is the date and references for the statement that CHF, however, is typically associated with impaired natriuretic peptide signaling, i. e. with a natriuretic peptide resistance, in part because of their enhanced internalization and degradation via NPR3? Old literature described desensitization of NPR1, but I am not aware of high qualities studies on NPR3 in this regard.

Page 16 line 401 correct mores to more.

1. Miyazaki T, Otani K, Chiba A, Nishimura H, Tokudome T, Takano-Watanabe H, et al. A New Secretory Peptide of Natriuretic Peptide Family, Osteocrin, Suppresses the Progression of Congestive Heart Failure After Myocardial Infarction. *Circ Res.* Mar 2018;122(5):742-51. Epub 2018/01/11.

Reviewer #2 (Remarks to the Author):

In this manuscript, Kattih et al performed RNA sequencing analysis of wasting skeletal muscle, and found that the expression levels of Musclin, a skeletal muscle derived myokine, were significantly

decreased during pressure overload-induced heart failure. Skeletal muscle specific disruption of Musclin in mice exaggerated the progression of pressure overload-induced heart failure, while AAV6-mediated overexpression of Musclin in skeletal muscle attenuated cardiac dysfunction and fibrosis. The authors further claimed that Musclin exerted its cardioprotective effects by enhancing C-type natriuretic peptide (CNP) signaling in cardiomyocytes.

This study represents a large body of work, and the results obtained from loss-of-function and gain-of-function experiments in vivo are interesting. However, molecular mechanisms underlying the cardioprotective effects of Musclin proposed by the authors are not convincing.

- 1) In Fig.1, plasma levels of Musclin should be measured in WT-sham and WT-TAC mice.
- 2) If the authors would like to claim that Musclin exerts its biological effects via the NP clearance receptor (NPR3), competition of CNP binding to NPR3 by Musclin should be demonstrated. Previous paper cited by the authors only showed the competition of ANP binding to NPR3 by Musclin (Kita S et al J Endocrinol 2009).
- 3) Although myocardial levels of CNP were increased, plasma levels of ANP were not significantly increased in AAV6-Musclin treated mice (Fig.3q and Supplementary Fig.2r). Plausible explanation for this discrepancy should be provided.
- 4) As the authors state in the manuscript, Musclin is expressed almost exclusively in skeletal muscle. However, skeletal muscle specific disruption of Musclin only resulted in minor reduction of Musclin plasma levels (Fig.4g). Plausible explanation should be provided.
- 5) Myocardial levels of CNP and plasma levels of ANP should be measured in WT and Musclin KO mice in parallel.
- 6) The authors examined pharmacological effects of Musclin in vitro using isolated cardiomyocytes and cardiac fibroblasts (Fig.6 -Fig.8). The authors should examine the CNP levels in the culture supernatant before and after Musclin addition in order to confirm if Musclin efficiently blocks the CNP clearance by NPR3.
- 7) It should be tested whether the absence of the NP clearance system affects pharmacological effects of Musclin in vitro by knocking out or knocking down NPR3 expressed in cultured cardiomyocytes and cardiac fibroblasts.

Reviewer #3 (Remarks to the Author):

The manuscript by Kattih et al. is timely and well written. It aims to elucidate the role of the skeletal muscle secreted myokine musclin on cardiac function and myocardial structural changes in a TAC model of cardiac cachexia. As cardiac cachexia is a major complication of heart failure and the mechanisms of its development are not fully understood, this is a very important research question and the methods used are state of the art. However, there are some key issues that need to be addressed.

1) The authors claim that their animal model is a model of cardiac cachexia, yet the only reference to body weight is found in figure 1f, that show the body weight at the end of the 12 week study period. The hallmark symptom of cachexia is non-edematous weight loss. Therefore, the authors need to show the change in body weight from baseline to week 12 and not just the final weight. Ideally, they show weight/ growth curves. This is also necessary for the AAV6 Co and AAV6 Muc animals as well as the Osn KO or overexpression animals. If the authors are unable to provide such data, they should not claim that this model is a cardiac cachexia model and omit the cachexia term and instead use the term heart failure model.

2) Figure 1, the number for the analyses were 3 and 4 respectively and yet a Students t-test was used to analyze the data. That is not an adequate test for such small number and a non-parametric test like Mann–Whitney U test should be used, as you absolute cannot expect the data to have a Gaussian distribution, on which the t-test is based. Hence, the data need to be re-analyzed or the number of animals increased.

3) Figure 3, again low number, Gaussian distribution has to be analyzed before selecting a relevant statistical test.

4) Figure 1k, how do the authors explain the loss in triceps (and to a lesser extend the quadriceps) weight in the sham group?

5) In the methods section the authors state: “No statistical method was used to predetermine sample size”. This is somewhat surprising as the authors have a wealth of experience in the TAC model, which must allow them to perform a sample size calculation. This would have been important for the experiments.

6) Fat wasting is an important part of cachexia and yet no data on fat is given. A cross-talk of fat and muscle has been shown (Das et al., Science. 2011 Jul 8;333(6039):233-8.) Ideally, the body composition of the animals would have been analyzed sequentially during the experiments.

POINT BY POINT RESPONSES TO REVIEWER'S COMMENTS

We would like to thank the reviewers for their thoughtful comments that helped us to improve our manuscript.

Below the reviewer comments appear in bold, followed by our response in normal font.

Reviewer #1:

The article titled, “Skeletal muscle derived Musclin protects the heart during pathological overload,” by Kattih et al, describes murine studies investigating the protective role of Musclin in cardiac dysfunction. They found that Musclin knockout mice had exaggerated cardiac dysfunction while mice overexpressing of Musclin in muscle had attenuated cardiac dysfunction. More specifically, they performed TAC on mice to pressure overload the heart for 12 weeks and saw reduce leg muscle weights consistent with cachexia. They isolated RNA from the leg muscle of heart failed or control mice and performed RNAseq analysis. They found that about 800 genes were upregulated and about 800 genes were downregulated. One of the mRNAs that was downregulated was Musclin, which was downregulated 3.5-fold.

They also determined that Musclin levels were decreased by 30% in human heart failure patients compared to healthy individuals, although this small decrease seems unlikely to have an impact on Npr1 and Npr2 signaling.

Musclin plasma levels in our knock-out mice are also only reduced by about 30%, but we still found a significantly reduced cardiac function, more fibrosis and significant reductions in cardiomyocyte ANP and CNP as well as cGMP and cAMP levels in Musclin knock-out mice (see revised/new Figures 4o, Figures 7a-b). It could also well be that Musclin levels in the plasma compartment do not well represent its tissue levels, where it is exerting its main effects.

When Musclin was knockout, heart failure consequences were exacerbated, but when Musclin was overexpressed by an AAV vector, heart failure was ameliorated. This is reasonable work, but some of the cardiac results have been previously reported (1). However, the most important new information is “how” the increased musclin levels improve heart function. Is it through Npr1, Npr2 or possibly Npr3? Unfortunately, the experiments that are designed to differentiate between these pathways rely on unproven inhibitors of Npr1 and Npr2. A much better approach is to selectively remove these receptors in specific tissues using Cre/Lox technology and ask if the effects go away, or at the very least, are markedly diminished, in mice where these receptors are not expressed in the tissues hypothesized to be critical for the physiologic effects of Musclin. Floxed mice for both Npr1 and Npr2 have been available for years.

This is a good point. We now use cardiomyocytes of cardiomyocyte specific *Npr1* knock-out mice as well as of *Npr2* knock-out mice to analyze the effects of Musclin on cellular cAMP, cGMP as well as on cellular contractility. In this regard, we added the new Figures 5h-j, Figures 6e-h and Supplementary Figure 5e-f. To address the role of Musclin in cardiac fibroblasts, we used specific siRNAs to downregulate NPR-A, B or C in these cells. The new results of these experiments are shown in Figure 8c-h and Supplementary Figure 6a-c.

Major issues:

A large amount of the data supporting this manuscript is based on so called selective inhibitors of Npr1, Npr2 and/or PKG. In this reviewer's opinion, there are no specific inhibitors of Npr1, Npr2 or Npr3. If the authors disagree, then please show data where Npr1 or Npr2 were expressed in NP receptor lacking cells that have been transfected with a single NP receptor, which is required to show which receptor is being activated. In this situation, is there data showing that an inhibitor inhibits one receptor and not the other? I am not aware of papers that report these critical data so please reference it. A much better and believable way to conduct a similar experiment is to use mice with Lox P sites flanking the genes for Npr1 or Npr2 in order to specifically delete these receptors in specific cell types.

We now use cardiomyocytes of cardiomyocyte specific *Npr1* knock-out mice as well as of *Npr2* knock-out (KO) mice to analyze the effects of Musclin on cellular cAMP, cGMP as well as on contractility. We found that Musclin (in the presence of CNP) enhances cardiomyocyte contractility (measured as cell shortening) in cardiomyocytes of wild-type and of *Npr1* KO mice, but not in *Npr2* KO mice (new Figures 5h-j). This indicated that NPR-B is necessary for the contractility enhancing effects of Musclin in the presence of CNP. In line with these data, we found that Musclin (in the presence of CNP) enhances the generation of cGMP and of cAMP in WT and *Npr1* KO cardiomyocytes, but not in *Npr2* KO cardiomyocytes, suggesting that NPR-B is responsible for Musclin effects on cGMP and cAMP in cardiomyocytes (new Figures 6e-h and Supplementary Figure 5e-f).

Unfortunately, we were not able to use adult cardiomyocytes lacking NPR-C. Therefore, we added a note of caution in the discussion of the revised manuscript (page 18):

“Importantly, there is strong evidence emerging from Npr3 knock-out mice that direct NPR-C dependent signaling (for example involving Gi proteins and ERK signaling) could be responsible for the protective roles of CNP^{58, 59, 60}. In our study, however, Musclin/CNP co-stimulation triggered increased cAMP levels in cardiomyocytes (as opposed to decreased cAMP levels that would have been expected due to NPR-C-Gi signaling) and also did not affect ERK signaling in cardiac fibroblasts. Moreover, a vital cardioprotective role was recently demonstrated for the CNP receptor NPR-B, as heterozygous Npr2 knock-out mice develop heart failure with myocardial fibrosis in advanced age⁶¹. This, together with the abrogation of Musclin effects in cardiomyocytes and fibroblasts by NPR-B ablation/downregulation, argues against a major role of direct NPR-C signaling in mediating the effects of Musclin, although our study was not optimally designed to clarify this point, which will need further investigation in the future. In addition, it is possible that Musclin acts (at least in part) via a currently unknown (co-) receptor.”

To address the role of Musclin in cardiac fibroblasts, we used specific siRNA to downregulate NPR-A, B or C in these cells. We found that fibroblast proliferation and migration were markedly suppressed by Musclin, CNP and a combination of both in siControl, siNpr3 as well as siNpr1 treated fibroblasts, but neither Musclin, CNP nor their combination had an effect on the migration or proliferation of siNpr2 treated fibroblasts (new Figures 8c-h). Therefore, also in fibroblasts, Musclin acts via CNP/NPR-B signaling.

If the authors hypothesis is correct, the AAV directed expression of Musclin should dramatically increase ANP, BNP and CNP levels in these mice. Is this true?

In the revised manuscript, we demonstrate now that in agreement with our hypothesis, skeletal muscle overexpression of Musclin (by AAV6) markedly increased ANP as well CNP levels in the myocardium after TAC surgery (please see new Figures 3q). In turn, Musclin ablation in skeletal muscle in the KO mice exerted reduced ANP and CNP levels in specific

time points after TAC, again in support of our hypothesis that Musclin blocks the degradation of the natriuretic peptides. Unfortunately, we did not find a way to measure BNP levels, but we feel that our measurements of ANP and CNP nicely support our hypothesis.

In addition, an increase/decrease in cardiomyocyte cGMP levels in response to increased or decreased skeletal muscle Musclin levels (see revised/new Figures 7a-d) support our conclusions.

Furthermore, we conducted an NPR-C competition assay in Hek293 cells with or without transfection of NPR-C-GFP. In this assay NPR-C transfected Hek cells had lower CNP levels in their supernatant, while supplementation of Musclin increased CNP levels in the supernatant (please see the new Figure 5b-c in the revised manuscript).

Minor Comments:

It makes no sense to say that “Musclin is expressed almost exclusively in skeletal muscle,” and in the very next phrase say that it is also expressed in bone. In fact, the same protein was identified by a separate group and named osteocrin.

We changed this in the introduction (page 4 of the revised manuscript) to: *“Musclin is expressed mainly in skeletal muscle and in a bit smaller amounts in bone (hence also referred to as osteocrin in the literature), but not in the heart”*,

As well as in the discussion (page 15) to: *“Musclin is expressed in skeletal muscle and in bone, where it was shown to mediate insulin-dependent glucose metabolism and endochondral growth, respectively^{34, 35, 36, 15, 16, 37, 38}”*

Thanks for indicating this mistake to us.

Musclin immunoblots are inconsistent and the differences look more like transfer issues rather than real differences in protein concentrations.

It took us quite a while in the process of this project to identify a good antibody to detect Musclin by Western blotting. However, with this antibody (#RD181079100, BioVendor) we are really confident that we detect a very specific signal to measure Musclin levels. For example, we do not find Musclin in the heart or in skeletal muscle of our knock-out mice, but we find increased levels after AAV6 mediated overexpression. We also confirmed the regulation of Musclin protein with immunofluorescence and at the RNA level in the different conditions. We show the uncropped blots in the source data files. In addition, we verified in the original blots that the protein transfer worked very well, please see the Figures below:

Fig. 1p

Fig. 3c

Fig. 4e

What is the date and references for the statement that CHF, however, is typically associated with impaired natriuretic peptide signaling, i. e. with a natriuretic peptide resistance, in part because of their enhanced internalization and degradation via NPR3? Old literature described desensitization of NPR1, but I am not aware of high qualities studies on NPR3 in this regard.

We softened our statement a little bit, to show that this is currently more a hypothesis. We also added references (page 5 of the revised manuscript):

“It is suggested that this occurs at least in part, because of enhanced internalization and degradation of natriuretic peptides via NPR-C^{22, 23}.”

References:

22. Kuhn M. Molecular Physiology of Membrane Guanylyl Cyclase Receptors. *Physiol Rev* 96, 751-804 (2016).

23. Andreassi MG, Del Ry S, Palmieri C, Clerico A, Biagini A, Giannessi D. Up-regulation of 'clearance' receptors in patients with chronic heart failure: a possible explanation for the resistance to biological effects of cardiac natriuretic hormones. *Eur J Heart Fail* 3, 407-414 (2001).

Page 16 line 401 correct mores to more.

We fixed this mistake. Thank you.

Reviewer #2:

In this manuscript, Kattih et al performed RNA sequencing analysis of wasting skeletal muscle, and found that the expression levels of Musclin, a skeletal muscle derived myokine, were significantly decreased during pressure overload-induced heart failure. Skeletal muscle specific disruption of Musclin in mice exaggerated the progression of pressure overload-induced heart failure, while AAV6-mediated overexpression of Musclin in skeletal muscle attenuated cardiac dysfunction and fibrosis. The authors further claimed that Musclin exerted its cardioprotective effects by enhancing C-type natriuretic peptide (CNP) signaling in cardiomyocytes.

This study represents a large body of work, and the results obtained from loss-of-function and gain-of-function experiments in vivo are interesting. However, molecular mechanisms underlying the cardioprotective effects of Musclin proposed by the authors are not convincing.

We now introduced a large amount of extra work to better understand how Musclin protects the heart. Among others, we now use cardiomyocytes of cardiomyocyte specific *Npr1* knock-out mice as well as of *Npr2* knock-out mice to analyze the effects of Musclin on cellular cAMP, cGMP as well as on contractility. In this regard, we added the new Figures 5h-j, Figures 6e-h and Supplementary Figures 5e-f. To address the role of Musclin in cardiac fibroblasts, we used specific siRNA to downregulate NPR-A, B or C in these cells. The new results of these experiments are shown in Figure 8c-h and Supplementary Figure 6a-c. We also showed that Musclin can increase the concentration of CNP in the supernatant of Hek293 cells transfected with NPR-C-GFP (new Figures 5b-c). In summary, these experiments suggested that a) Musclin acts in cardiomyocytes mainly via enhancement of

CNP/NPR-B signaling to increase their contractility and b) in fibroblasts also by enhancing CNP/ NPR-B signaling to reduce their migratory capacity and their proliferation.

1) In Fig.1, plasma levels of Musclin should be measured in WT-sham and WT-TAC mice.

We now did that and show the results in the new Figure 1r in the revised manuscript. As indicated there, Musclin levels are significantly reduced by about 50% in cachectic wild-type mice after long-term TAC.

2) If the authors would like to claim that Musclin exerts its biological effects via the NP clearance receptor (NPR3), competition of CNP binding to NPR3 by Musclin should be demonstrated. Previous paper cited by the authors only showed the competition of ANP binding to NPR3 by Musclin (Kita S et al J Endocrinol 2009).

To answer this important point we transfected Hek293 cells with a NPR-C-GFP expression plasmid or left the cells untransfected as control. We then added CNP +/- Musclin and measured CNP levels in the supernatant. We found that in the absence of Musclin, CNP levels tended to be lower in transfected cells versus untransfected cells, indicating that NPR-C degrades CNP. Addition of Musclin, however, significantly increased CNP levels in the supernatant of transfected cells (please see the new Figures 5b-c).

3) Although myocardial levels of CNP were increased, plasma levels of ANP were not significantly increased in AAV6-Musclin treated mice (Fig.3q and Supplementary Fig.2r). Plausible explanation for this discrepancy should be provided.

We found a strong tendency of increased ANP plasma levels in mice after AVV6 treatment (Supplemental Figure 2r), which due to the relatively high variability between samples did not reach statistical significance. Cardiomyocyte ANP levels, however, were increased by skeletal muscle specific overexpression of Musclin, but decreased in skeletal muscle Musclin KO mice after TAC. We hypothesize that Musclin mainly affects natriuretic peptide levels in their target organs (such as the heart), while the plasma levels are additionally dependent on other variables (such as renal excretion etc.).

4) As the authors state in the manuscript, Musclin is expressed almost exclusively in skeletal muscle. However, skeletal muscle specific disruption of Musclin only resulted in minor reduction of Musclin plasma levels (Fig.4g). Plausible explanation should be provided.

We measured Musclin expression in wild-type and our skeletal muscle specific knock-out mice in organs, where Musclin was previously reported to be expressed. As demonstrated in the new Figure 4h of our revised manuscript. Musclin mRNA expression is strongly reduced in quadriceps muscle of our KO mice, but normal Musclin expression is visible in bone (of the skull) and brain. Because in this assay bone Musclin expression is only modestly lower than in skeletal muscle we modified our statements about Musclin expression in the introduction: and

We modified our statement about Musclin expression in our introduction (page 4 of the revised manuscript) to: "*Musclin is expressed mainly in skeletal muscle and in a bit smaller amounts in bone (hence also referred to as osteocrin in the literature), but not in the heart*", as well as in the discussion (page 15) to: "*Musclin is expressed in skeletal muscle and in bone, where it was shown to mediate insulin-dependent glucose metabolism and endochondral growth, respectively* ^{34, 35, 36, 15, 16, 37, 38}."

Although Musclin levels are only mildly, but significantly reduced, the level of reduction is enough to exert functional effect during TAC, i.e the aggravation of cardiac dysfunction. The reduction of Musclin we observe in our Musclin KO mice is also similar to the one in patients with aortic stenosis (Figure 2).

5) Myocardial levels of CNP and plasma levels of ANP should be measured in WT and Musclin KO mice in parallel.

Since we cannot measure plasma CNP levels in a reliable manner and found ANP levels to be highly variable, we measured CNP as well as ANP levels in cardiomyocytes of mice after AAV6 mediated overexpression of Musclin in skeletal muscle, as well as after skeletal muscle specific Musclin KO. Musclin overexpression indeed increased cardiomyocyte ANP and associated CNP levels after TAC (new Figures 3q), while in Musclin KO, the opposite was observed: reduced myocardial ANP and CNP levels in selected time-points after TAC (new Figures 4o).

6) The authors examined pharmacological effects of Musclin in vitro using isolated cardiomyocytes and cardiac fibroblasts (Fig.6 -Fig.8). The authors should examine the CNP levels in the culture supernatant before and after Musclin addition in order to confirm if Musclin efficiently blocks the CNP clearance by NPR3.

We performed this important experiment in Hek293 cells with and without transfection with NPR-C-GFP and showed there that Musclin was able to increase CNP levels in the supernatant of the transfected Hek cells (new Figures 5b and c). With regard to the heart, we now show in the revised manuscript that Musclin overexpression or downregulation, increases or decreases cardiomyocyte cGMP (and cAMP levels, as result of the cGMP to cAMP crosstalk) levels, respectively (revised/new Figures 7a-d). We further show that the increase in cGMP (and cAMP) depend on the NPR-B receptor. With regard to isolated fibroblasts, we measured CNP levels in the supernatant at the end of the migration assay (i.e. 4 hours after stimulation of the cells). These results are therefore not comparable to the results from transfected/untransfected Hek cells, which were analyzed 1 hour after stimulation. Please see the diagram below for fibroblast supernatant CNP concentrations:

7) It should be tested whether the absence of the NP clearance system affects pharmacological effects of Musclin in vitro by knocking out or knocking down NPR3 expressed in cultured cardiomyocytes and cardiac fibroblasts.

This is also a great suggestion. We knocked down NPR-A, -B and -C in cardiac fibroblasts. While knocking down NPR-B ablated the effects of CNP and Musclin, knocking down NPR-A

or -C had only little effects (new Figures 8c-h). As NPR-C knock-down should (according to our hypothesis) have similar effects as the treatment with Musclin (i.e. the blockade of natriuretic peptide degradation), both intervention should have (if anything) agonistic effects. Indeed, Fibroblast proliferation during the stimulation with CNP, Musclin and the combination of both, appeared to be a little more effective in siNPR3(C) than in siControl treated fibroblasts. This effect was better visible in the migration assay, where Musclin and CNP exerted stronger inhibitory effects on cell migration in siNPR3(C) treated versus sicontrol treated fibroblasts.

Cellular contractility was determined in freshly isolated mouse adult cardiomyocytes, which cannot be cultured long enough to reach successful gene knock-down by siRNA. Therefore, we knocked down NPR-C in neonatal rat cardiomyocytes. In these cells, however, cellular contractility cannot be assessed, so we used phospholamban (PLB) phosphorylation as read out for the Musclin effects. As shown below, Musclin (Mu) and CNP enhanced phospholamban phosphorylation, which was even more exaggerated upon downregulation of NPR-C.

On the left, the downregulation of *Npr1-3* mRNA is shown after specific siRNA treatment, on the right an immunoblot for PLB, phospho-PLB and GAPDH (as loading control is shown):

Reviewer #3:

The manuscript by Kattih et al. is timely and well written. It aims to elucidate the role of the skeletal muscle secreted myokine musclin on cardiac function and myocardial structural changes in a TAC model of cardiac cachexia. As cardiac cachexia is a major complication of heart failure and the mechanisms of its development are not fully understood, this is a very important research question and the methods used are state of the art. However, there are some key issues that need to be addressed.

1) The authors claim that their animal model is a model of cardiac cachexia, yet the only reference to body weight is found in figure 1f, that show the body weight at the end of the 12 week study period. The hallmark symptom of cachexia is non-edematous weight loss. Therefore, the authors need to show the change in body weight from baseline to week 12 and not just the final weight. Ideally, they show weight/ growth curves. This is also necessary for the AAV6 Co and AAV6 Muc animals as well as the *Ostn* KO or overexpression animals. If the authors are unable to provide such data, they should not claim that this model is a cardiac cachexia model and omit the cachexia term and instead use the term heart failure model.

This is a good point. In the revised manuscript, we now show in the new Figure 1g the course of the body weight changes in wild-type mice after sham versus TAC surgery. As demonstrated in there, both groups of mice gain weight until 6 weeks after surgery, when the TAC mice start to lose weight decreasing back about to their starting weight. On the other hand, the sham mice continue to increase their body weight to almost double their starting weight. We also performed MRI in a cohort of mice. These examinations 12 weeks after surgery demonstrated a reduction inguinal and abdominal fat in TAC mice to about 1/3 of what can be observed in sham treated animals. In addition, a significant reduction in autochthonal muscle mass was also detected by this method in TAC mice. We also analyzed the reduction in muscle fiber size (in quadriceps muscle cross-sections) and found a significant reduction in fibers size after TAC versus sham mice. This demonstrated skeletal muscle wasting also on the histological level 12 weeks after TAC surgery.

We did not record many morphometric parameters in the cohort of mice presented in Figure 3, which was followed for 9 weeks after surgery. Therefore, we included a second cohort of mice treated with AAV6-Co or AAV-Musclin, which we had analyzed until 12 weeks after surgery, and where more morphometric parameters were recorded. As demonstrated in the **new Supplementary Figure 3** of the revised manuscript, HW/TL was markedly increased after TAC, but no difference was found upon AAV6-Musclin treatment. The lung weight/tibia length ratio was strongly increased as sign of pulmonary congestion and heart failure in AAV6-control treated mice, but this was completely blunted upon AAV6-Musclin administration, indicating better heart function in this group. With regard to echocardiographic systolic heart functional parameters, AAV6-Musclin improved this 9 and 12 weeks after TAC, although significance was only reached 9 weeks after surgery. Body weight was not different between the different groups before surgery, but decreased in AAV6-Control treated mice after TAC (versus sham) as one sign of cachexia (new Supplementary Figure 3f). With regard to skeletal muscle weights, TAC induced a reduction in quadriceps weight and the same trend was found in the gastrocnemius muscle in AAV-control treated mice. Interestingly, this was not visible in AAV6-Musclin treated mice, which even showed significantly increased gastrocnemius muscle weights versus AAV6-control treated mice after TAC. This might indicate that Musclin could ameliorate skeletal muscle wasting, although this was not the subject of this study.

We also show body and muscle weights in wild-type and Musclin knock-out (KO) mice 2 weeks after sham or TAC surgery (new Supplementary Figure 4m-o). However, at this early time point, we do not expect any features of cachexia after TAC. Accordingly, no significant changes in these parameters were found (new Supplementary Figure 4m-o).

The skeletal muscle specific KO mice were studied at this early time point after TAC, when endogenous Musclin levels are still unchanged, to see what effect a pure reduction of Musclin in skeletal muscle might have, without any associated changes of cachexia.

2) Figure 1, the number for the analyses were 3 and 4 respectively and yet a Students t-test was used to analyze the data. That is not an adequate test for such small number and a non-parametric test like Mann–Whitney U test should be used, as you absolute cannot expect the data to have a Gaussian distribution, on which the t-test is based. Hence, the data need to be re-analyzed or the number of animals increased.

To address this point, we operated new mice, and thereby increased the n of mice per group in new Figures 1b, c, e, f (formerly 1b-e).

Moreover, for all panels of our manuscript, we now tested for normal distribution with the Shapiro-Wilk normality test. If $n > 3$ and if normality was indicated by the test, we conducted a

parametric test (such as in the new Figure 1b-g), if this was not the case, such as in Figure 1o, but also in Figure 2 (human data), we conducted a non-parametric test: the Mann-Whitney U test in case two groups were compared and the Kruskal-Wallis test (with the Dunn's multiple comparison's post test), when more than two groups were compared.

We recalculated all p values accordingly for the revised manuscript. The statistical test that was used is mentioned in the Figure legends.

3) Figure 3, again low number, Gaussian distribution has to be analyzed before selecting a relevant statistical test.

As mentioned in the first answer, we used the Shapiro-Wilk Normality test for each panel. According to the results, we used either a parametric or a non-parametric test. We also show now a second cohort of this experiment in the new Supplementary Figure 3. So, altogether, quite a lot of mice were analyzed that had received either AAV6-control or AAV6-Musclin.

4) Figure 1k, how do the authors explain the loss in triceps (and to a lesser extend the quadriceps) weight in the sham group?

This is a misunderstanding. In this Figure, the Musclin (Ostn) mRNA expression (but not muscle weight) is analyzed at the different time points after sham or TAC surgery. We cannot explain, why Musclin mRNA expression levels decline over time in the Triceps muscle. Since Musclin expression is also regulated by muscle activity (Subbotina et al., 2015, PNAS), it could be that the mice move less with increasing age within in the cage. However, this is only a hypothesis, and most importantly, Musclin expression was still significantly lower after TAC.

5) In the methods section the authors state: "No statistical method was used to predetermine sample size". This is somewhat surprising as the authors have a wealth of experience in the TAC model, which must allow them to perform a sample size calculation. This would have been important for the experiments.

We chose sample size based on our experience in the TAC model with regard to the variability of the parameters (such as echocardiographic ejection fraction) we were interested in. However, we appreciate this advice by the reviewer and will pre-calculate sample size in future studies.

6) Fat wasting is an important part of cachexia and yet no data on fat is given. A cross-talk of fat and muscle has been shown (Das et al., Science. 2011 Jul 8;333(6039):233-8.) Ideally, the body composition of the animals would have been analyzed sequentially during the experiments.

This is indeed a very interesting point. It could well be that factors from wasting fat tissue also impacts Musclin expression in skeletal muscle, although fat itself does not express Musclin. We measured the amount of fat by MRI 12 weeks after sham or TAC surgery and found a strongly reduced amount of inguinal and abdominal fat (new Figure 1 h-k). Because mice after TAC are quite sensitive to the lengthy MRI procedure, we only analyzed one time point at the end of the study with this method, but we show multiple time points for the body-weight analysis (new Figure 1g).

REVIEWER COMMENTS

Reviewer #1 (Remarks to the Author):

The manuscript is improved by the additional experiments and resulting data, but now with over 20 authors, it has become difficult to digest in one reading. Nonetheless, this reviewer has no significant issues with the current manuscript. However, attention to the two small issues below with further improve it.

1. References 20 and 22 are the same.

2. Panel q of figure 3 seems odd. CNP is only 22 amino acid, so how does one blot for such a small peptide? What is the percentage of the acrylamide in the gel? How do we know that the corresponding band is actually CNP? Is there a blocking peptide control? I have been in this field for a long time and cannot recall seeing a “western blot” for CNP, which makes one wonder why I am seeing one now.

Reviewer #2 (Remarks to the Author):

Skeletal muscle derived Musclin protects the heart during pathological overload

The authors performed a large amount of extra work to better understand the molecular mechanisms underlying the cardioprotective effects of Musclin. However, two critical concerns pointed out by this reviewer have not been addressed satisfactorily.

1) If the authors would like to claim that Musclin exerts its biological effects via the NP clearance receptor (NPR3), competition of CNP binding to NPR3 by Musclin should be demonstrated. Previous paper cited by the authors only showed the competition of ANP binding to NPR3 by Musclin (Kita S et al J Endocrinol 2009).

2) It should be tested whether the absence of the NP clearance system affects pharmacological effects of Musclin in vitro by knocking out or knocking down NPR3 expressed in cultured cardiomyocytes and cardiac fibroblasts.

To address the first concern, the authors demonstrated new Fig. 5b and c. Although new Fig. 5b and c are consistent with the authors' hypothesis, the designed experiments are indirect, and the results are weak. The direct evidence of Musclin binding to NPR3 should be provided by the standard biochemical binding assay.

To address the second concern, the authors provided new Fig. 8c-h. Surprisingly, NPR3 (NPR-C) knock-down had not abolished the pharmacological effects of Musclin, which is totally opposite to the author' hypothesis.

Thus, the molecular mechanisms proposed by the authors are still in doubt.

Reviewer #4 (Remarks to the Author):

The authors generally provide adequate answers to most of the raised concerns and they have added some substantial data which is appreciated. They attempt to provide novel data on an important area and the lack of knowledge between skeletal muscle-cardiac cross talk is an important and interesting issue.

The case for a true state of cachexia are still rather weak and was not consistent across treatments, which still raises some doubts in what is the appropriate term to use here (e.g., perhaps muscle wasting is better). In this regard, however, at present muscle weights are normalized to tibia length, given the loss of body weight it would be more appropriate to provide both absolute wet weights and normalised for body weight, which would fully confirm a state of muscle wasting. I suspect once normalised to body weight (not tibia length) no differences in muscle weights will be apparent. Some histology are presented to show some mild forms of muscle atrophy but this in only one muscle group and not even fibre-type specific, limiting interpretation. It would be useful to also assess various muscle groups and stain in a fibre-specific manner. As the authors eluded, Musclin can be impacted by physical activity and we also know cachexia is driven by poor nutrition. It would have been useful to rule out whether differences in physical activity and food intake were driving some of the major effects observed between CON vs TAC, which we known play a key role in cachexia/wasting.

Beyond this a couple of important issues still require clarification.

1) An important issue using cardiac surgery is to confirm that all mice in the treatment groups start from the same baseline. Following TAC it would have been important to confirm, say after 1 week using echo, that all mice after the surgery were well matched with the same degree of cardiac dysfunction, thus ruling any effects of surgical differences per se on disease severity. It seems this was first done at 3 weeks, but it would have been useful to confirm at the baseline starting point otherwise any differences between treatment groups could be caused by surgical differences alone. In this regard, the most strong evidence would be if the authors found evidence Musclin treatment after the development of overt heart failure was able to reverse cardiac dysfunction. This is the most clinically relevant situation.

2) Fig. 2 of the patient data is not convincing given controls are 20 years younger. Therefore the difference in Musclin could simply be caused by aging and not cardiac disease. In addition no data are given if any patients have diabetes which can impact Musclin (Chen et al., 2017 Diabetes & Vascular Disease Res 14: 116-). Incidentally this study find the opposite and Musclin concentrations are higher following T2D development.

POINT BY POINT RESPONSES TO REVIEWER'S COMMENTS

We would like to thank the reviewers for their thoughtful comments that helped us to improve our manuscript.

Below the reviewer comments appear in bold, followed by our response in normal font.

The manuscript is improved by the additional experiments and resulting data, but now with over 20 authors, it has become difficult to digest in one reading. Nonetheless, this reviewer has no significant issues with the current manuscript. However, attention to the two small issues below will further improve it.

1. References 20 and 22 are the same.

Thanks for noticing this. We removed the duplicate of this reference.

2. Panel q of figure 3 seems odd. CNP is only 22 amino acid, so how does one blot for such a small peptide? What is the percentage of the acrylamide in the gel? How do we know that the corresponding band is actually CNP? Is there a blocking peptide control? I have been in this field for a long time and cannot recall seeing a “western blot” for CNP, which makes one wonder why I am seeing one now.

Thank you for pointing this out. Indeed, the blot that we demonstrate in Figure 3q (for the situation of AAV control and AAV Musclin) as well as for Figure 4o (for wild-type and Musclin KO) show pro-ANP and pro-CNP (both around 15 kDa). We clarify this now in the revised manuscript. We found that both pro-ANP and pro-CNP increase after TAC in control/WT mice 9 weeks (Fig. 3q) as well as 2 weeks (Fig. 4o) after TAC. The response after 9 weeks was enhanced by AAV-Musclin administration, while the response after 2 weeks was reduced in Musclin KO versus wild-type. We ran those blots on a 15% SDS-page gel. Using the same antibody as we did in our study, Konstanze Michel, Michaela Kuhn et al. (JCI insight, 2020, Figure 2b) have also demonstrated increased pro-CNP levels in the left ventricle after TAC by Western blot. In addition, we have used a blocking peptide and indeed found a strongly reduced signal in the respective Western blot, indicating the specificity of our approach (see Figure below). Pro-CNP (amino acid 1-103) has been found recently in plasma as the predominant form together with CNP-53 (Wilson MO et al., Peptides, 2018). Because higher molecular weight forms of CNP (like pro-CNP) exert slower rates of degradation, while more mature CNP peptides are extremely short-lived, pro-CNP might be a good readout to demonstrate Musclin dependent effects on CNP degradation. In addition, using a commercial ELISA, detecting total mature CNP peptides (CNP-22, CNP-29 and CNP-52), we demonstrate reduced plasma CNP levels in Musclin KO versus wild-type mice after TAC (new Figure 4p) in the revised manuscript.

Figure 3q

Full unedited Western Blot membrane incubated with anti-CNP (Thermo Fischer Scientific #PA5-47454)

Full unedited Western Blot membrane stripped and incubated with anti-CNP [1 µg/ml], which was pre-incubated with recombinant CNP [10 µg/ml]

Figure 4o

Full unedited Western Blot membrane incubated with anti-CNP (Thermo Fischer Scientific #PA5-47454)

Full unedited Western Blot membrane stripped and incubated with anti-CNP [1 µg/ml], which was pre-incubated with recombinant CNP [10 µg/ml]

Reviewer #2:

Skeletal muscle derived Musclin protects the heart during pathological overload

The authors performed a large amount of extra work to better understand the molecular mechanisms underlying the cardioprotective effects of Musclin. However, two critical concerns pointed out by this reviewer have not been addressed satisfactorily.

1) If the authors would like to claim that Musclin exerts its biological effects via the NP clearance receptor (NPR3), competition of CNP binding to NPR3 by Musclin should be demonstrated. Previous paper cited by the authors only showed the competition of ANP binding to NPR3 by Musclin (Kita S et al J Endocrinol 2009).

2) It should be tested whether the absence of the NP clearance system affects pharmacological effects of Musclin in vitro by knocking out or knocking down NPR3 expressed in cultured cardiomyocytes and cardiac fibroblasts.

To address the first concern, the authors demonstrated new Fig. 5b and c. Although new Fig. 5b and c are consistent with the authors' hypothesis, the designed experiments are indirect, and the results are weak. The direct evidence of Musclin binding to NPR3 should be provided by the standard biochemical binding assay.

We now included this biochemical binding assay in the new Figure 5 b) and c) in the revised manuscript, by which we show that Musclin binds to NPR-C, but can be displaced by increasing amounts of recombinant CNP. The following passage has been added to the Results section in the revised manuscript (page 12):

“To address whether Musclin in principal can compete with CNP for its binding to NPR-C, we designed an ELISA based assay, in which we coated plastic wells with NPR-C and applied Musclin together with increasing concentrations of CNP. Musclin that bound to NPR-C in the well was subsequently detected by an ELISA method (Figure 5b). As shown in Figure 5c, increasing concentrations of CNP were able to prevent binding of Musclin to NPR-C in the well and thereby led to Musclin displacement, indicating that indeed CNP and Musclin compete for binding NPR-C.”

To address the second concern, the authors provided new Fig. 8c-h. Surprisingly, NPR3 (NPR-C) knock-down had not abolished the pharmacological effects of Musclin, which is totally opposite to the author' hypothesis.

We demonstrate in Figure 8 that downregulation of *Npr2* (NPR-B) by siRNA abrogates the pharmacological effects of Musclin on cardiac fibroblast proliferation and migration. This is in accordance with our hypothesis that Musclin increases CNP levels and thereby enhances NPR-2 (NPR-B) dependent signaling. Downregulation of *Npr3* by siRNA, in contrast, even slightly enhances the Musclin dependent effects, especially on fibroblast migration. This is in line with our hypothesis. We (together with previous data, see for example reference 19 in our manuscript) propose that Musclin acts by blocking NPR-C dependent degradation of natriuretic peptides. Therefore, downregulation of NPR-C by siRNA should have a similar effect as Musclin itself and should promote its effects as long as the effect has not reached a maximum.

Thus, the molecular mechanisms proposed by the authors are still in doubt.

Sorry, for explaining our results not enough. We have added the following paragraph to the Results section of our revised manuscript (Page 15/16):

“The inhibitory effects of Musclin and CNP on fibroblast proliferation and migration were abolished by downregulation of NPR-B via siRNA, but were essentially unchanged by siRNA mediated downregulation of NPR-A, or by treatment with control siRNA (Fig. 8c-h, Suppl. Fig. 8a-b). This indicated that sufficient expression of NPR-B, but not of NPR-A was necessary for Musclin and/or CNP mediated inhibition of fibroblast activity. Downregulation of NPR-C also did not ablate and even partially enhanced the effects of CNP and Musclin, especially with regard to cell migration (Fig. 8c-h, Suppl. Fig. 8c). In this case, downregulation of NPR-C had a similar effect as Musclin, which blocks NPR-C, and therefore both maneuvers act agonistically.”

Reviewer #4 (Remarks to the Author):

The authors generally provide adequate answers to most of the raised concerns and they have added some substantial data which is appreciated. They attempt to provide novel data on an important area and the lack of knowledge between skeletal muscle-cardiac cross talk is an important and interesting issue.

We want to thank this new reviewer for this statement in support of our work.

The case for a true state of cachexia are still rather weak and was not consistent across treatments, which still raises some doubts in what is the appropriate term to use here (e.g., perhaps muscle wasting is better). In this regard, however, at present muscle weights are normalized to tibia length, given the loss of body weight it would be more appropriate to provide both absolute wet weights and normalised for body weight, which would fully confirm a state of muscle wasting. I suspect once normalised to body weight (not tibia length) no differences in muscle weights will be apparent. Some histology are presented to show some mild forms of muscle atrophy but this in only one muscle group and not even fibre-type specific, limiting interpretation. It would be useful to also assess various muscle groups and stain in a fibre-specific manner.

We kindly disagree with this reviewer on this point. Former reviewer #3 stated that *“The hallmark symptom of cachexia is non-edematous weight loss. Therefore, the authors need to show the change in body weight from baseline to week 12 and not just the final weight.”* This is in line with the definition of cachexia in humans, according to the Evans criteria, in which a reduction in body weight (due to fat and muscle loss like in our study) is a major criterion for the existence of cachexia (see Reference #26 in our manuscript). Unfortunately, cachexia has not yet been defined with similar clarity in mice. During the first revision we had provided a body weight time course and show a reduction in body weight in TAC mice from 24.6g (6 weeks after TAC) to 21.78 g (12 weeks after TAC). This amounts to a weight loss of 12% in 6 weeks. In contrast, the sham operated mice gained 4g of weight in the same period (+15.2%). Overall, the sham operated mice were 39% heavier than the TAC operated mice at the end of the study. In addition, we demonstrated a reduction in muscle weight previously only in quadriceps and gastrocnemius muscle. We now added triceps brachii, plantaris and soleus muscles, which all showed a reduction of their total mass after TAC (please see new Figure 1e-i). In addition, we demonstrated diminished autochthonal muscle mass in MRI and now in the revised manuscript also a reduction of the psoas muscle (please see new Figure 1m) after TAC. We not only show muscle wasting after TAC, but also the reduction in inguinal fat mass (weighing of fat) as well as reduced abdominal fat (in MRI). Furthermore, we now performed ATPase staining on quadriceps muscle after TAC in the revised manuscript and re-quantified the reduction in muscle fiber areas. As previously described for a different heart failure model (Li P et al., Am J Pathol, 2007), mainly the glycolytic IIb and IIc fibers exerted a reduced fiber area (please see new Figures 1p-r) in our study after TAC. In addition, we now conducted qPCRs from quadriceps, gastrocnemius and triceps brachii muscle 12 weeks after sham or TAC surgery and found typical cachexia associated genes, which includes wasting regulated genes (Atrogin 1, MuRF-1, FOXO-1), but also the pro-inflammatory gene TNF-alpha (also known as cachectin), upregulated 12 weeks after TAC (please see new Supplementary Figure 1f-g).

We feel that all data together makes a strong case for the presence of cachexia after long-term TAC. It is also not just muscle wasting, since we see major reductions in fat mass and cachexia triggered gene expression. In the following 4 high impact papers, cachexia was

indicated by exactly the same measures as we took (reduction in body weight, muscle weight and reduced muscle cross sectional area in one muscle):

-Zhou X et al., Reversal of Cancer cachexia and Muscle Wasting by ActRIIB Antagonism Leads to Prolonged Survival, *Cell*, 2010

-Fukawa T et al., Excessive fatty acid oxidation induces muscle atrophy in cancer cachexia, *Nature Medicine*, 2016

-Harper S.C., GDF11 Decreases Pressure Overload-Induced Hypertrophy, but Can Cause Severe Cachexia and Premature Death, *Circulation Research*, 2018

-Klose R. et al., Targeting VEGF-A in myeloid cells enhances natural killer cell responses to chemotherapy and ameliorated cachexia, *Nature Communications*, 2016

Skeletal muscle weight in mouse cachexia studies is indeed mostly shown “unnormalized” (just in mg), but also normalized to tibia length. Since the body weight goes down in cachexia, in large parts due to the reduction in muscle weight, normalization to body weight will certainly often ablate the differences between cachectic and non-cachectic conditions. Due to the suggestion of this reviewer, we now show unnormalized muscle weights in main Figure 1, but muscle weight/body weight ratios in the new Supplementary Figure 1a-e. As shown there and also in here below, despite normalization to body weight, we still found a significantly reduced quadriceps, gastrocnemius and triceps brachii weight during TAC, but the differences were ablated for plantaris and soleus muscles.

The reviewer also stated that the development of cachexia was not consistent between treatments. Indeed, AAV-Musclin treated animals developed less cachexia 12 weeks after TAC, but they started out at the same weight (see Supplemental Figure 5f). This indicated that Musclin has even some anti-cachectic properties in addition to its beneficial effects on the heart, which makes the protein even more interesting.

As the authors eluded, Musclin can be impacted by physical activity and we also know cachexia is driven by poor nutrition. It would have been useful to rule out whether differences in physical activity and food intake were driving some of the major effects observed between CON vs TAC, which we known play a key role in cachexia/wasting.

To measure physical activity and nutrition would have indeed been interesting, but goes beyond what our manuscript aims to do (which is to address the muscle-heart axis in heart failure for the first time). If one wanted to decipher the mechanism of cachexia in heart failure, these measures would clearly need to be addressed, but this was not the goal of our study.

Beyond this a couple of important issues still require clarification.

1) An important issue using cardiac surgery is to confirm that all mice in the treatment groups start from the same baseline. Following TAC it would have been important to confirm, say after 1 week using echo, that all mice after the surgery were well matched with the same degree of cardiac dysfunction, thus ruling any effects of surgical differences per se on disease severity. It seems this was first done at 3 weeks, but it would have been useful to confirm at the baseline starting point otherwise any differences between treatment groups could be caused by surgical differences alone. In this regard, the most strong evidence would be if the authors found evidence Musclin treatment after the development of overt heart failure was able to reverse cardiac dysfunction. This is the most clinically relevant situation.

To address this point, another cohort of mice was analyzed (termed “second cohort”, please see new Supplementary Figure 4). This cohort was treated exactly like the previously analyzed cohorts (Figure 3, Supplementary Figure 5), but we conducted in addition a Doppler analysis of right versus left carotid artery blood flow 2 days after surgery, as well as earlier echocardiographic analyses. We found that the degree of pressure overload as indicated by Doppler analysis was similar between AAV Control and AAV Musclin treated groups. In addition, cardiac systolic function (ejection fraction) was still similar between both groups 1 week after TAC (and thus 5 days after application of the AAV6 vectors), but became significantly improved by AAV Musclin after 3 weeks of TAC. This data shows that indeed Musclin dependent improvement of heart function emerges after treatment with AAV Musclin.

We have also taken a clinically relevant approach, because AAV6 treatment in all cohorts analyzed in this study started two days after pressure overload had been established. Since protein expression from AAV6 transduced tissue needs a few days to emerge, effective treatment probably started around 5-7 days after the beginning of pressure overload.

2) Fig. 2 of the patient data is not convincing given controls are 20 years younger. Therefore the difference in Musclin could simply be caused by aging and not cardiac disease. In addition no data are given if any patients have diabetes which can impact Musclin (Chen et al., 2017 Diabetes & Vascular Disease Res 14: 116-). Incidentally this study find the opposite and Musclin concentrations are higher following T2D development.

The reviewer is right that the patient cohort in our basic science study was indeed not perfectly matched with regard to age. Although no data are available that Musclin serum levels change between the age of 60 and 80 years, we do acknowledge this now clearly as limitation of our analysis (page 7 of the revised manuscript): “These patients were on average older than the healthy individuals, which is a limitation of this analysis (Table 2).”

The aspect of diabetes mellitus is interesting. We found that 8 of the analyzed patients suffered from diabetes mellitus (this is now also indicated in Table 2 in the revised manuscript). When we excluded these diabetic patients, the differences between Musclin levels became even more significant (see new Figure 2b vs. 2c).

We are very excited that we now got the chance to analyze skeletal muscle biopsy tissues (vastus lateralis muscle) from control individuals as well as from patients with heart failure and sarcopenia or cachexia in collaboration with Stephan von Haehling, Jochen Springer and Jens Fielitz. As indicated in the new Figure 2a, cachectic heart failure patients exerted a strong and significant downregulation of *OSTN* (encoding for Musclin) mRNA expression in skeletal muscle, while sarcopenic heart failure patients also had markedly reduced *OSTN* mRNA levels, but this did not reach statistical significance. Clinical data of the patients and controls are demonstrated in the new Table 1. These new findings highlight the potential relevance of our study for patients with cardiac cachexia.

REVIEWER COMMENTS

Reviewer #1 (Remarks to the Author):

One odd aspect of this paper is that they are examining proCNP and not CNP-22. The latter is the biologically relevant species. In fact, some researchers use proCNP levels as a control when evaluating the degradation of CNP-22. Hence, it does not seem appropriate to use proCNP as a surrogate for the biological, e.g. signaling, form of CNP. Presumably, the whole point of the paper is to show that changes in musclin compete with ANP, BNP and CNP-22 for degradation by NPR-C, NEP as well as other peptidases, which ultimately elevates the smaller, signaling forms of each natriuretic peptide. My recollection is that the "pro" versions of these peptides are not degraded by the same pathways as the biologically relevant processed molecules. In fact, several groups, have reported that adding just a few amino acids to the ends of these molecules dramatically reduces their degradation. Hence, in this reviewer's opinion, only the levels of the smaller signaling forms of the peptides are relevant to this story. Measuring levels of CNP-22 is critical to accurately evaluating the validity of this hypothesis.

Reviewer #2 (Remarks to the Author):

This reviewer understands the authors' hypothesis that Musclin increases CNP levels and enhances NPR-B signaling by directly blocking NPR-C dependent degradation of CNP. Thus, downregulation of NPR-C by siRNA should have similar effects as those of Musclin administration, and the pharmacological effects of CNP should be enhanced in the absence of NPR3.

In the right half of the Figs. 8e and h (siNPR3-treated cells), a significant difference was observed between the fourth bar from the right (vehicle+, CNP-, Musclin-) and the third bar from the right (vehicle+, CNP+, Musclin-), which is consistent with the hypothesis described above. However, in the same figures (the right half of the Figs. 8e and h: siNPR3-treated cells), a significant difference was still observed between the fourth bar from the right (vehicle+, CNP-, Musclin-) and the second bar from the right (vehicle+, CNP-, Musclin+), suggesting that Musclin exerts its pharmacological effects even in the absence of NPR3. This finding is totally opposite to the authors' claim that NPR-C mediates the pharmacological effects of Musclin.

Reviewer #4 (Remarks to the Author):

The authors have done an excellent job addressing my concerns and have added not only relevant, but important additional clinical data, to strengthen the study.

POINT BY POINT RESPONSES TO REVIEWER'S COMMENTS

Below the reviewer comments appear in bold, followed by our response in normal font.

Reviewer #1 (Remarks to the Author):

One odd aspect of this paper is that they are examining proCNP and not CNP-22. The latter is the biologically relevant species. In fact, some researchers use proCNP levels as a control when evaluating the degradation of CNP-22. Hence, it does not seem appropriate to use proCNP as a surrogate for the biological, e.g. signaling, form of CNP. Presumably, the whole point of the paper is to show that changes in musclin compete with ANP, BNP and CNP-22 for degradation by NPR-C, NEP as well as other peptidases, which ultimately elevates the smaller, signaling forms of each natriuretic peptide. My recollection is that the "pro" versions of these peptides are not degraded by the same pathways as the biologically relevant processed molecules. In fact, several groups, have reported that adding just a few amino acids to the ends of these molecules dramatically reduces their degradation. Hence, in this reviewer's opinion, only the levels of the smaller signaling forms of the peptides are relevant to this story. Measuring levels of CNP-22 is critical to accurately evaluating the validity of this hypothesis.

We want to thank this reviewer for her/his valuable comment. Although we consistently saw increased levels of pro-CNP and pro-ANP in the myocardium of AAV6-Musclin treated mice vs. AAV6-control treated mice after TAC, we now removed this piece of data and eliminated the Western blot from Figure 3q. We agree with this reviewer that the mechanism how pro-CNP (and pro-ANP) increase in the myocardium upon overexpression are not clear yet. We instead now measured *mature* CNP peptides (CNP-22, CNP-29, CNP-52) in the plasma of mice treated with AAV6-Control or AAV6-Musclin 21 days after TAC (as shown in Supplementary Figure 4A). As now demonstrated in the new Figure 3q, AAV6-Musclin treated mice exert significantly **higher levels of mature CNP** after TAC in plasma (compared to AAV6-control treated mice), likely due to decreased degradation of mature CNP, when Musclin levels are increased. Mature CNP levels in plasma were assessed with a CNP ELISA kit (#MBS2514795, detecting mature CNP peptides, MyBioSource, also used for Figure 4o and Figure 5e).

In addition, we removed Western blots showing consistently higher levels of pro-CNP and pro-ANP in the myocardium of wild-type versus skeletal muscle specific Musclin knock-out mice after TAC from former Figure 4o for the same reasons. We instead now only demonstrate **reduced levels of mature CNP peptides** in the plasma of skeletal muscle specific Musclin knock-out mice after TAC (versus wild-type), which indicates increased CNP degradation upon skeletal muscle Musclin knock-out.

These data together with data from Figure 5b-e, which show competition of CNP and Musclin for the NPR3 support our hypothesis that Musclin acts (at least in part) by increasing CNP levels. Ablation of Musclin/CNP or Musclin dependent effects upon knock-out/knock-down of NPR2 in Figure 5l (cardiomyocytes) and Figure 8d and g (fibroblasts) shows the functional relevance of CNP/NPR2 for the impact of Musclin.

Reviewer #2 (Remarks to the Author):

This reviewer understands the authors' hypothesis that Musclin increases CNP levels and enhances NPR-B signaling by directly blocking NPR-C dependent degradation of CNP. Thus, downregulation of NPR-C by siRNA should have similar effects as those of Musclin administration, and the pharmacological effects of CNP should be enhanced in the absence of NPR3.

In the right half of the Figs. 8e and h (siNPR3-treated cells), a significant difference was observed between the fourth bar from the right (vehicle+, CNP-, Musclin-) and the third bar from the right (vehicle+, CNP+, Musclin-), which is consistent with the hypothesis described above. However, in the same figures (the right half of the Figs. 8e and h: siNPR3-treated cells), a significant difference was still observed between the fourth bar from the right (vehicle+, CNP-, Musclin-) and the second bar from the right (vehicle+, CNP-, Musclin+), suggesting that Musclin exerts its pharmacological effects even in the absence of NPR3. This finding is totally opposite to the authors' claim that NPR-C mediates the pharmacological effects of Musclin.

We want to thank this reviewer for her/his valuable comment, although we respectfully disagree with the argumentation in the second paragraph. Indeed, administration of Musclin is still able to inhibit fibroblast migration and proliferation upon knocking down NPR3 (even in a somewhat enhanced manner as compared to siControl treated fibroblasts, although this does not reach statistical significance). We explain this by the fact that we have done only a knock-down (i.e. incomplete deletion, see also Supplementary Figure 8) of NPR3 and not a knock-out (complete deletion of NPR3). The knock-down of NPR3 leads to a reduction in NPR3 levels. A reduced number of NPR3 molecules can now be more efficiently bound and blocked by a given concentration of Musclin compared to a situation where more NPR3 is present (and still the same amount of Musclin). Therefore, in the case of the combination of siNPR3 and Musclin we have a "double" inhibition, since both siNPR3 as well as Musclin inhibit NPR3 dependent effects, which is stronger than if we only have a single inhibition when only Musclin, but no siNPR3 is there. We apologize for expressing this point not clearly in previous versions of our manuscript. We have now explained this point better on page 16 of the revised manuscript: *"In this case, partial downregulation of NPR-C by siRNA had a similar effect as Musclin, which blocks NPR-C, and therefore both maneuvers act in the same direction to antagonize NPR-C."*

In the case of siNPR2 the case is a little different: here, Musclin and/or CNP can no longer inhibit fibroblast proliferation and migration. Knock-down of NPR2 reduces the level of NPR2 in fibroblasts. For a given concentration of CNP, not enough NPR2 receptor is present to mediate functional effects in these cells. Here we have only a "single inhibition" during agonistic stimulation, and not a double inhibition (like in the case of siNPR3 and Musclin), because CNP is an agonist at the NPR2 receptor (while Musclin, of course, is proposed to be an antagonist at the NPR3 receptor).

Reviewer #4 (Remarks to the Author):

The authors have done an excellent job addressing my concerns and have added not only relevant, but important additional clinical data, to strengthen the study.

We want to thank this reviewer for her/his positive comments and his constructive suggestions that helped us to improve our work and now even add clinical data.

REVIEWERS' COMMENTS

Reviewer #1 (Remarks to the Author):

The additional data showing that CNP-22 is increased in response to NPR3 and musclin has satisfied my concerns about the mechanism of musclin. I have no additional issues.

Reviewer #2 (Remarks to the Author):

NCOMMS-19-10578C

Skeletal muscle derived Musclin protects the heart during pathological overload

The authors claim that Musclin increases CNP levels and enhances NPR2 signaling by directly blocking NPR3 dependent degradation of CNP. Thus, it is critical to examine whether the absence of NPR3 abolishes the pharmacological effects of Musclin by performing loss-of-function experiments.

In this manuscript, loss of NPR3 function experiments were only provided in Fig. 8 utilizing siNPR3 treatment. Upon siNPR3 treatment, Musclin still exerted its pharmacological effects, and the authors state that this is because knock-down of NPR3 was incomplete and a reduced number of NPR3 were still expressed. This reviewer understands that incomplete deletion of NPR3 leads to enhanced NPR2 signaling, and the authors' hypothesis that a reduced number of NPR3 may be able to mediate the effects of Musclin.

However, this incomplete deletion of NPR3 performed by the authors is a flawed loss-of-function experiment, and therefore does not address the most critical concern that whether the effects of Musclin are really mediated by NPR3. If siRNA treatment cannot knock-down NPR3 sufficiently, the authors should knock-out NPR3 and clarify whether the complete loss of NPR3 affects the pharmacological activity of Musclin.

Point by point response to reviewer #2:

Please find the reviewer's points in **bold**, and our response in normal font.

Reviewer #2 (Remarks to the Author):

NCOMMS-19-10578C

Skeletal muscle derived Musclin protects the heart during pathological overload

The authors claim that Musclin increases CNP levels and enhances NPR2 signaling by directly blocking NPR3 dependent degradation of CNP. Thus, it is critical to examine whether the absence of NPR3 abolishes the pharmacological effects of Musclin by performing loss-of-function experiments.

Our claim that Musclin increases CNP levels by directly blocking NPR3 is based on previous literature: Kita S et al., (2009, Reference #19 in our manuscript) have shown that Musclin binds to NPR3 and that it can inhibit the binding of ANP to this receptor. No binding of Musclin was found to NPR1 or NPR2. Miyazaki et al. (2018, Reference #17 in our manuscript) demonstrated that NPR3 knock-out mice do not show the reduction in blood pressure in response to Musclin as it is evident in wild-type mice, in principal showing that NPR3 is needed for Musclin dependent effects. Perhaps even more importantly, Kanai Y et al. (2017, Reference #16 in our manuscript) demonstrated that systemic transgenic Musclin expression led to increased circulating CNP levels and enhanced bone growth. This enhanced bone growth by increased Musclin levels was ablated by a CNP knock-out as well as by a NPR3 knock-out. This indicates that Musclin can induce functional effects through CNP and through NPR3. In the light of this existing data, elucidating further the role of NPR3 for Musclin dependent effects was not our highest priority.

The novelty in our paper is rather:

- a) That Musclin is downregulated in skeletal muscle (in mice and human patients) suffering from cardiac cachexia;
- b) to establish the cardiac consequences of skeletal muscle specific overexpression as well as skeletal muscle specific knock-out of Musclin, showing that muscle derived Musclin enhances cardiomyocyte contractility and inhibits cardiac fibroblast activity and fibrosis;
- c) to show that the protective effects of Musclin in the heart depend on NPR2, because Npr2 knock-out cardiomyocytes do not show enhanced cardiomyocyte contractility and Npr2 knock-down fibroblasts do not become activated in response to Musclin. We further show increased and decreased cardiac cGMP and cAMP levels in response to skeletal muscle over-expression and downregulation of Musclin, respectively. We therefore show for the first time a skeletal muscle-heart crosstalk, which is critically relevant in heart failure.

We added a lot of data with regard to the role of NPR3 for Musclin effects on the heart (mainly in response to this reviewer: the competition of CNP and Musclin for binding to NPR-3, increased CNP levels in cell culture in response to Musclin addition, increased/decreased serum CNP levels in response to overexpression and knock-out of Musclin in skeletal

muscle, respectively). In our opinion, these data (together with the published data mentioned above) are a strong surrogate for the importance of NPR3 in mediating Musclin effects in the heart. However, we agree with the reviewer that it will be important to work on the role of NPR3 for Musclin dependent cardiac effects in the future, especially in the in vivo situation using Npr3-knock-out mice, which we acknowledge in our revised manuscript (Discussion, page 20): *“The role of NPR-C for the cardiac Musclin effects in vivo should be evaluated in Npr3 knock-out mice in the future.”*

In this manuscript, loss of NPR3 function experiments were only provided in Fig. 8 utilizing siNPR3 treatment. Upon siNPR3 treatment, Musclin still exerted its pharmacological effects, and the authors state that this is because knock-down of NPR3 was incomplete and a reduced number of NPR3 were still expressed. This reviewer understands that incomplete deletion of NPR3 leads to enhanced NPR2 signaling, and the authors’ hypothesis that a reduced number of NPR3 may be able to mediate the effects of Musclin. However, this incomplete deletion of NPR3 performed by the authors is a flawed loss-of-function experiment, and therefore does not address the most critical concern that whether the effects of Musclin are really mediated by NPR3. If siRNA treatment cannot knock-down NPR3 sufficiently, the authors should knock-out NPR3 and clarify whether the complete loss of NPR3 affects the pharmacological activity of Musclin.

We agree with this reviewer that a full knock-out of NPR3 would have been better than an incomplete knock-down. We still feel, however, that our knock-down results give important clues: it shows that Npr3 knock-down and Musclin act in the same direction. We acknowledge this in the revised manuscript (page 16, Results): *“Although this gives a hint on the role of NPR-C for Musclin dependent effects, a complete ablation/knock-out of NPR-C would have given a better clue in this regard.”* We also feel that the strong dependency of the cardiac Musclin effects on NPR2 presence speaks clearly for the mechanism that we are proposing.